# KANK1 promotes breast cancer development by compromising Scribble-mediated Hippo activation

Shiny Shengzhen Guo [1] ✉, Zhiying Liu[2], Guan M. Wang [1], Zhiqi Sun [1], Kaikai Yu[1], James P. Fawcett[3], Reinhard Buettner [4], Bo Gao [2,5] & Reinhard Fässler [1]

KANK1 is expressed in epithelial cells and connects focal adhesions with the adjacent cortical microtubule stabilizing complex. Although KANK1 was shown to suppress cancer cell growth in vitro, TCGA database points to high KANK1 levels associated with poor prognosis in a wide spectrum of human malignancies. Here, we address this discrepancy and report that KANK1 promotes proliferation and survival of PyMT-transformed mammary tumor cells in vivo. Mechanistically, KANK1 localizes to the basal side of basement membrane (BM)-attached transformed luminal epithelial cells. When these cells lose the contact with the BM and disassemble integrin adhesions, KANK1 is found at cell-cell junctions where it competes with the polarity and tumor suppressor Scribble for NOS1AP binding, which curbs the ability of Scribble to promote Hippo pathway activity. The consequences are stabilization and nuclear accumulation of TAZ, growth and survival of tumor cells and elevated breast cancer development.

Integrin-mediated cell adhesion to the extracellular matrix (ECM) is essential for metazoan life. A hallmark of integrins is that upon ECM binding, they cluster and assemble large signaling hubs called focal adhesions (FAs)[1]. The Kidney Ankyrin Repeat-containing Proteins (KANKs) are a family of FA proteins that are recruited to the outer rim of FAs via a direct interaction between the unique KANK-N-terminal (KANK-KN) motif and Talin[2–5]. Two of the KANKs, KANK1 and KANK2 were studied in more detail and shown to be also present in the cortical microtubule stabilizing complex (CMSC)[2,3], which assembles in the close vicinity of FAs. The recruitment of KANK1 and KANK2 to the CMSC is mediated by the direct binding of the KANK-coiled-coil domains (KANK-CC) and the five ankyrin repeats (KANK-ANKs) to the CMSC proteins Liprinβ1 and KIF21A[4–6]. Although it is not known whether KANK1 and KANK2 physically link CMSCs and FAs, they functionally link FAs and CMSCs in two ways. On one hand, KANK1 and KANK2 regulate cell mechanics and FA dynamics by modulating the

actomyosin linkage with Talin and the release of the RhoA-activating GEF-H1 from microtubules (MTs). On the other hand, they promote ECM turnover through the secretion of cargo such as metalloproteinases (MMPs) by capturing MT tracks next to FAs[4,7–9].

Although the four KANK proteins share motif and domain structure, they are expressed in a cell type-specific manner pointing to cell- and tissue-specific functions. KANK1 is primarily expressed in epithelial cells, KANK2 and KANK4 in mesenchymal cells and KANK3 in endothelial cells[10]. Assessments by medical genetics identified germline mutations of the *KANK* genes associated with rare human congenital disorders, including cerebral palsy (*KANK1*), nephrotic syndrome (*KANK1*, *KANK2* and *KANK4*) and keratoderma (*KANK2*)[11–13]. Methylation of the promoter/exon1 region in the human *KANK1* gene leading to loss or decreased KANK1 expression is associated with renal cell carcinoma (RCC)[14]. The latter finding, together with observations that overexpression of KANK1 decreases proliferation of intestinal as well as

[1]Department of Molecular Medicine, Max Planck Institute of Biochemistry, Martinsried, Germany. [2]School of Biomedical Sciences, LKS Faculty of Medicine, The University of Hong Kong, Hong Kong SAR, China. [3]Departments of Pharmacology and Surgery, Dalhousie University, Halifax, NS B3H 4R2, Canada. [4]Institute of Pathology, Medical Faculty, University Cologne, Cologne, Germany. [5]School of Biomedical Sciences, Faculty of Medicine, The Chinese University of Hong Kong, Hong Kong SAR, China. ✉e-mail: shguo@biochem.mpg.de

breast cancer cell lines through cell-cycle arrest or induction of apoptosis, suggests that KANK1 might function as a tumor suppressor[15–22]. In contrast to these observations, however, The Cancer Genome Atlas (TCGA) database indicates that high rather than low levels of KANK1 expression are associated with poor prognosis in a variety of human malignancies including breast cancer[23–25] (Fig. 1a–f). Based on this finding, we decided to resolve this contradiction by defining and characterizing the tumor activity of KANK1 in breast cancer using different model systems.

The activity and kinetics of integrin adhesions to the ECM play a major role in the outcome of a malignancy. A key for this function is that integrins and several FA proteins are mechanosensitive, which enables integrin adhesions to sense the biophysical property of the ECM, transmit the information and activate mechanosensitive signaling pathways such as FAK, Src and YAP/TAZ[1,26–28]. The YAP/TAZ signaling molecules are paralog transcriptional co-activators that associate with DNA-bound TEAD factors to induce the transcription of genes that regulate cancer cell stemness, proliferation and migration[29–31]. Besides mechanotransduction, YAP/TAZ activity can also be regulated by the Hippo kinase cascade, which dictates whether YAP/TAZ is sequestered in the cytoplasm, degraded by the ubiquitin-proteasome system (UPS) or shuttled into the nucleus[32,33]. Extracellular signals that activate the Hippo pathway are not known. However, the protocadherin FAT1 and the polarity protein SCRIB (Scribble) can efficiently induce Hippo activation[29,34,35]. SCRIB localizes to the basolateral membrane where it assembles a protein complex with the nitric oxide synthase 1 adaptor protein (NOS1AP)[36], YAP/TAZ and the Hippo kinases LATS and MST leading to YAP/TAZ phosphorylation and degradation. Conversely, the loss of SCRIB from the basolateral membrane, which is a hallmark of the EMT-induced polarity loss, releases YAP/TAZ from MST/LATS inhibition, leading to their nuclear translocation[37–41].

NOS1AP, also known as CAPON (carboxyl-terminal PDZ ligand of neuronal nitric oxide synthase), exists in numerous complexes. NOS1AP can bind and activate dexamethasone-induced Ras-related protein 1 (Dexras1) leading to enhanced neuronal nitric oxide synthase (nNOS) signaling[42], synapsins I-III leading to nNOS localization to pre-synaptic terminals[43], and SCRIB leading to increased dendritic spine formation in cultured hippocampal neurons[44] and inhibition of YAP activity in the breast cancer cell line MCF7[36]. The NOS1AP gene is spliced into different isoforms, which have unique tissue and sub-cellular distributions. Only NOS1AP isoforms containing the amino-terminal phospho-tyrosine binding (PTB) domain are recruited to the plasma membrane and associate with SCRIB[36]. Genetic studies implicated the NOS1AP locus in schizophrenia[45], QT syndrome[46] and nephrotic syndrome[47] in human. Whether and how NOS1AP influences breast cancer in vivo is not known.

In the present study, we find that KANK1 promotes tumor growth by changing the subcellular location from integrin adhesions to cell-cell junctions, where KANK1 competes with SCRIB for the binding to NOS1AP. The KANK1-induced disruption of the SCRIB/NOS1AP complex curbs TAZ inhibition, resulting in increased mouse tumor cell growth in vivo, in tumoroids and in xenografted human cancer cells.

## Results

### KANK1 promotes mammary gland tumor growth

In vitro studies with cancer cell lines of different origin suggest that KANK1 functions as a tumor suppressor[14,16–22]. To decipher the mechanism of KANK1's role in tumor development in vivo, we generated the Kank1-floxed allele in mice (Supplementary Fig. 1a). We used a Cre-deletor transgene[48] to constitutively delete exon 6 and 7 of the Kank1 gene. This approach generates premature stop codons in exon 9 leading to non-sense mediated mRNA decay and loss of KANK1 translation products in different organs of 21- and 24-weeks old females, which was confirmed with an antibody recognizing peptide sequences encoded by exon 5 (Supplementary Fig. 1b). Mice carrying the disrupted Kank1 gene were termed KANK1-KO mice and back-crossed to C57BL/6. Offspring of heterozygous intercrosses were born at normal Mendelian ratios, displayed no obvious developmental and behavioral defects and lived a normal life span.

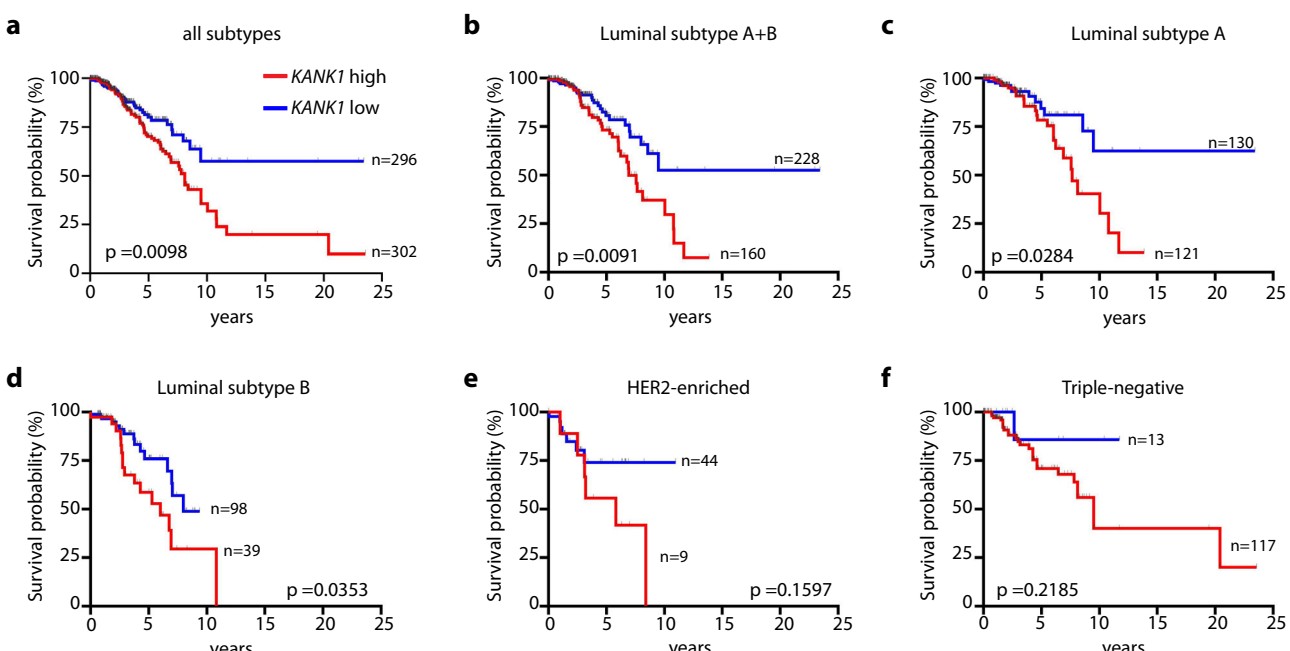

**Fig. 1 | Correlation of KANK1 gene expression with overall survival in human breast cancer. a–f** Kaplan-Meier (KM) survival analysis of patients from the TCGA cohort with high (red line) versus low (blue line) KANK1 expression. In terms of KANK1 gene expression, patients were divided into high and low risk groups based on a quartile risk model. The patient cohort analyzed in Fig. 1a was further stratified into 4 different subtypes based on the molecular characteristics. A second round of KM survival analysis was performed after subtyping (**b–f**). The number of patients included in the analysis is indicated on each graph. P values were calculated using the Log-rank test. Source Data are provided as a Source Data file.

Next, we used the KANK1-KO mouse strain to assess the tumor function of KANK1 in a breast cancer model. We favored breast cancer for our studies because Kaplan-Meier survival analysis of the TCGA breast cancer cohort revealed that high, rather than the expected low KANK1 expression correlates with a poor survival rate of breast cancer patients. The results of the in silico analysis points to an oncogenic role of KANK1 in breast cancer (Fig. 1a), which is in sharp contrast to the proposed tumor suppressor function of KANK1 observed with *in cellulo* studies[14,16–22]. Stratification of the TCGA breast cancer cohort based on their molecular characteristics revealed that poor prognosis correlates with high *KANK1* mRNA and protein levels in all subtypes, irrespective whether they are ER+ or of luminal subtype A or B (Fig. 1b–f, Supplementary Fig. 1c). Furthermore, we found that KANK1, whose expression is absent from virgin and involuted mammary glands, induced during pregnancy and lactation exclusively in luminal epithelial cells (LECs) (Supplementary Fig. 2a, b), was not compensated by de novo expression of KANK2-4 in KANK1-null glands (Supplementary Fig. 2c–e). Other cells in the mammary gland including the smooth muscle actin-positive (SMA+) contractile myoepithelial cells surrounding the glands express KANK2 and KANK4 (Supplementary Fig. 2d, e). Finally, the LEC polarity in active mammary glands (Supplementary Fig. 2a), the gland morphology in pregnancy, lactation, and involution (Supplementary Fig. 3a) as well as the offspring numbers were normal in

KANK1-null females (Supplementary Fig. 3b), indicating that mammary glands were unaltered prior to breast cancer induction.

To decipher the potential oncogenic function of KANK1, we induced breast cancer by crossing KANK1-WT and KANK1-KO mice with the polyoma middle-T (PyMT) mouse luminal breast cancer model that closely recapitulates human breast cancer development and progression from hyperplasia and early carcinoma in situ to invasive carcinoma[49]. In this model, the oncogenic transformation is induced by the transgenic expression of the oncoprotein PyMT in mammary gland LECs[49,50]. Whereas all KANK1-WT mice intercrossed with the PyMT transgene (KANK1-WT^PyMT) developed palpable tumors after a latency period of 8-16 weeks, only 50% of the KANK1-KO^PyMT mice developed palpable tumors after 16 weeks (Fig. 2a). At the 16-week time the KANK1-WT^PyMT tumor burden was three-fold higher (Fig. 2b) and the tumor size significantly larger compared to KANK1-KO^PyMT tumors (Fig. 2c, Supplementary Fig. 4a, b). At the 21-week timepoint, most of the KANK1-WT^PyMT tumors reached the volume and weight approved by the ethical committee, whereas the KANK1-KO^PyMT tumor weights were comparable to those of the KANK1-WT^PyMT tumors measured at the 16-week time point (Fig. 2b). In line with the accelerated tumor growth, hematoxylin/eosin (H/E) sections of KANK1-WT^PyMT tumors showed numerous small foci of carcinoma in situ at initial palpation, which increased in number and size at 16 weeks, and additionally, began to breach the basement membrane (BM) and invade the

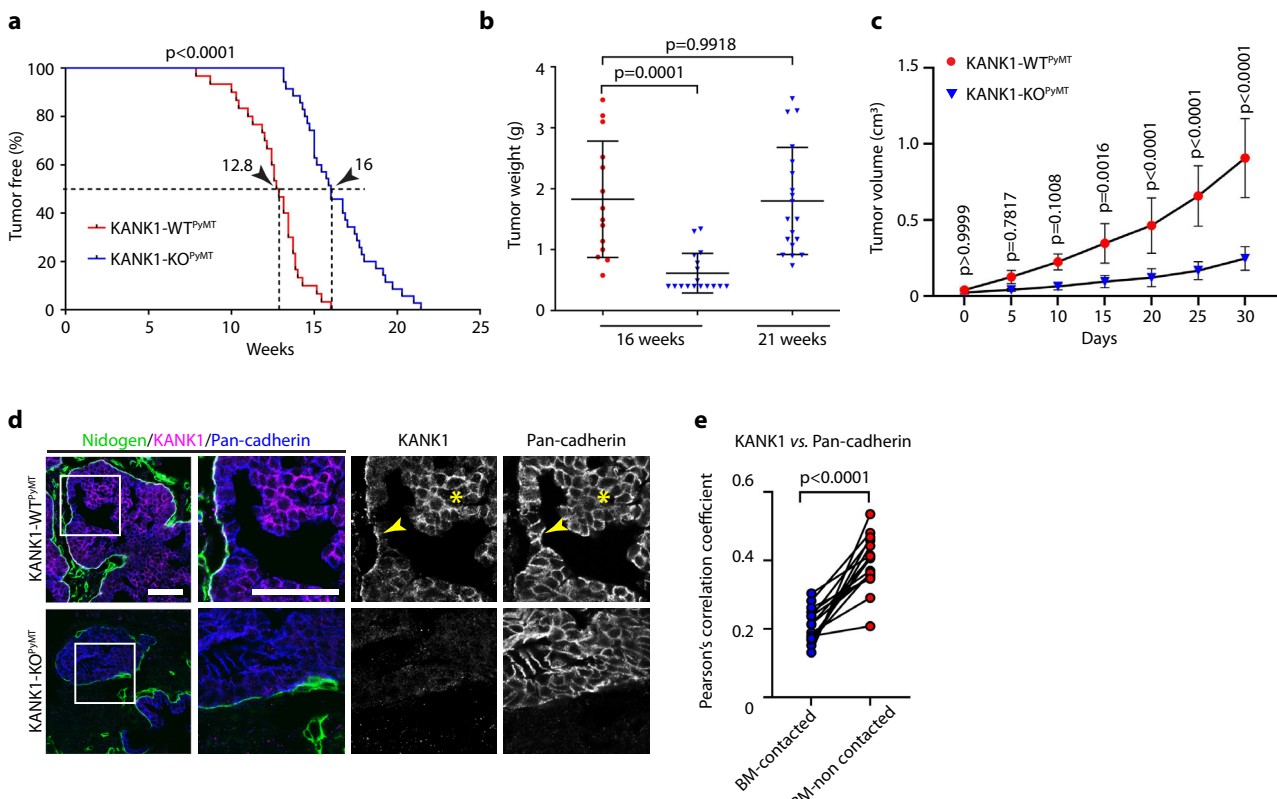

**Fig. 2 | KANK1 promotes mammary gland tumor development. a** Tumor incidence shown as tumor-free survival curve for KANK1-WT^PyMT (*n* = 30) and KANK1-KO^PyMT (*n* = 35) mice. Arrow heads indicate medium tumor latency. *P* values were calculated by Log-rank test. **b** Weight of tumors in KANK1-WT^PyMT (red) and KANK1-KO^PyMT (blue) mice at the experimental end points of 16 weeks (*n* = 14 for KANK1-WT^PyMT, and 17 for KANK1-KO^PyMT) and 21 weeks (*n* = 19 for KANK1-KO^PyMT). *P* values (one-way ANOVA with Dunnett's multiple comparisons test) are indicated on top of each comparison. Mean ± SD is shown. **c** Tumor growth rate between KANK1-WT^PyMT and KANK1-KO^PyMT littermates (*n* = 6 pairs). Tumor volume after initial palpation was monitored every 5 days for 30 days (see also Supplementary Fig. 4a for tumor growth curve within the same time window). *P* values were calculated using

two-way ANOVA with Šídák's multiple comparisons test and indicated for each comparison. Mean ± SD is shown. **d** Carcinoma in situ stained for KANK1 (magenta), Nidogen for basement membranes (green) and Pan-cadherin for cell-cell contacts (blue). KANK1 localizes at basal side (arrow heads) and cell-cell junctions (asterisks) of tumor cells. Tumor tissues were obtained from 16-week-old littermates and KANK1-KO^PyMT tumor tissues served as KANK1 antibody staining control (*n* = 8 independent repeats). Scale bars: 50 μm. **e** Pearson's correlation coefficient analysis of the co-localization between KANK1 and Pan-cadherin in the BM-non contacted and BM-contacted tumor cells. Data were collected from 16 regions across 5 images. *P* values (paired Student's *t* test, two-tailed) are indicated on top. Source Data are provided as a Source Data file.

surrounding tissue (Supplementary Fig. 4c, d). KANK1-KO[PyMT] mice developed fewer numbers of small foci of carcinoma in situ scattered throughout the otherwise normal mammary gland at the time of initial tumor palpation. These foci also increased in size and number, however, rarely progressed to invasive carcinoma at 16 weeks after tumor palpation (Supplementary Fig. 4c, d).

To identify the cell(s) that express KANK1 in KANK1-WT[PyMT] tumors, we immunostained tumor sections and observed high KANK1 levels exclusively in KANK1-WT[PyMT] LECs (Fig. 2d). The KANK1 protein switched from the predominantly basal side of tumor cells that were in contact with the BM (arrowhead in Fig. 2d) to the pan-cadherin+ cell cortex of tumor cells that were growing with no contact to the BM (asterisk in Fig. 2d). Expectedly, KANK1 was absent in KANK1-KO[PyMT] tumors. Immunostaining of tumor tissues harvested from virgin KANK1-WT[PyMT] mice at different tumor stages revealed that non-transformed LECs lacked KANK1 expression (Supplementary Fig. 4e), whereas transformed, single-layered hyperplastic LECs exhibited KANK1 expression at their basal side and weak KANK1 signals at cell-cell junctions (arrowheads in Supplementary Fig. 4e), and LECs of multilayered carcinomas showed strong KANK1 signals at cell-cell junctions (Fig. 2e, asterisks in Fig. 2d and Supplementary Fig. 4e). This finding implicates that tumor cell transformation and stratification rather than the expression of the PyMT oncogene per se serves as trigger for KANK1 induction. To further exclude stratification as trigger for the de novo expression of KANK1, we investigated whether KANK1 is expressed during development in terminal end buds (TEBs), which are composed of two distinct epithelial cell types, stratified E-cad+ body cells surrounded by the single-layered SMA+ cap cells. The latter cells represent progenitor cells that generate both, myoepithelial and luminal cells[51–53]. Our immunostaining clearly revealed that KANK1 is only expressed at the basal side of cap cells located at the TEB front, and neither in the TEB neck region nor in the multiple-layered body cells as well as differentiated myoepithelial cells (Supplementary Fig. 4f, arrow heads).

## KANK1 promotes tumor cell survival and proliferation

The aggravated tumor formation in KANK1-WT[PyMT] mice could result from increased proliferation and/or survival of transformed LECs. Immunostaining of phosphorylated histone H3 (pH3) and cyclin D1 revealed a significantly increased number of proliferating cells in KANK1-WT[PyMT] tumors (Fig. 3a, Supplementary Fig. 5a), and immunostaining of cleaved caspase-3 (clCasp3) and the TUNEL assay revealed a 3·5-fold decrease of apoptotic cells in KANK1-WT[PyMT] compared to KANK1-KO[PyMT] tumors (Fig. 3b, Supplementary Fig. 5b).

Since KANK1 was only expressed in PyMT-transformed tumor cells and absent in myoepithelial cells and tumor stromal cells including fibroblasts, endothelial cells and infiltrating immune cells (Supplementary Fig. 5c–f), we concluded that the tumor growth promoting effect of KANK1 is likely cell autonomous. To confirm this hypothesis, we isolated cytokeratin 8 (CK8)-positive and vimentin-negative LECs from mouse tumors, cultured them for 5 days in Matrigel and monitored the growth of the forming tumoroids by live-imaging. In line with the in vivo findings, KANK1 was enriched at the basal side of cells attached to the BM and at cell-cell contacts of cells away from the BM (Fig. 3c). The KANK1-WT[PyMT] tumoroids grew faster compared to KANK1-KO[PyMT] tumoroids (Fig. 3d, e, Supplementary Movie 1, 2) and contained significantly more Ki67+ cells at the experimental end point (Fig. 3d, f), which confirms our hypothesis that KANK1 promotes tumor growth in a cell-autonomous manner.

## KANK1 binds NOS1AP at cell-cell junctions

To unravel how KANK1 is recruited to the pan-cadherin+ cell-cell junctions, we searched for KANK1 interacting proteins in the human MCF7 breast cancer cell line, which expressed KANK1 at FAs and cell-cell junctions (Fig. 4a) and are non-invasive and hence, resemble primary PyMT-induced carcinoma in situ tumor cells[54,55]. To identify KANK1 interactors, we expressed green fluorescent protein (GFP)-tagged KANK1 (KANK1-GFP) or GFP-only in MCF7 cells and performed mass spectrometry (MS) on anti-GFP immunoprecipitates. The experiments identified the known KANK1 binding partners such as Talin1, DYNLL2, PPFIBP1 (Liprinβ1), KIF21A and 14-3-3 proteins (YWHAB, YWHAQ, YWHAZ and YWHAE)[2–6] and two additional interactors, nitric oxide synthase 1 adaptor protein (NOS1AP) and the NOS1AP-splice isoform chromosome 1 open reading frame 226 (C1orf226) (Fig. 4b). The interaction of KANK1 with NOS1AP and C1orf226 was also confirmed by MS in HEK293T cells (Supplementary Fig. 6a).

The NOS1AP protein family consists of several alternatively spliced isoforms (NOS1APa-c; Supplementary Fig. 6b), including canonical isoforms lacking C1orf226 and non-canonical isoforms containing C1orf226[36]. By gene cloning and sequencing the canonical as well as non-canonical NOS1AP isoforms were found to be expressed in MCF7 cells. qRT-PCR measurements with primers amplifying the PTB-domaining containing NOS1AP isoforms revealed high expression of mRNA expression of the non-canonical NOS1APc and low expression of canonical NOS1APa in MCF7 cells (Supplementary Fig. 6c, d). To test whether the NOS1AP interaction is shared by all KANK paralogues, we immunoprecipitated transiently overexpressed GFP-tagged KANK1-4 in MCF7 cell lysates and probed the immunoprecipitates with a commercial anti-panNOS1AP antibody that recognizes all NOS1AP isoforms. The identity of the KANK1 signal was determined by staining the Western blot (WB) with anti-GFP as well as anti-KANK1 antibodies (Supplementary Fig. 6e). The experiments revealed that KANK1 and to a much lower extent KANK2 co-precipitated a - 90 kD NOS1AP isoform, which corresponds to the non-canonical NOS1APc isoform (Fig. 4c, d).

Conversely, expression and measurement of the non-canonical GFP-NOS1APc interactome in MCF7 cells by MS revealed known NOS1AP interactors such as SCRIB and VANGL1 and in addition, KANK1 in the co-immunoprecipitate (Supplementary Fig. 6f). Importantly, immunoprecipitation of endogenous KANK1 in MCF7 cell lysates also co-precipitated the endogenous -90 kDa NOS1APc isoform (Fig. 4e, f), which confirms the interaction also between the two endogenous proteins.

To identify the regions that mediate binding between KANK1 or NOS1AP, we expressed truncation and deletion mutants of KANK1 and NOS1APc in HEK293T cells and performed co-immunoprecipitation experiments (Fig. 5a–f). The KANK1 mutants tagged with a C-terminal GFP included the KN motif, the N-terminal unstructured region (UN), the coiled-coil domain (CC), the C-terminal unstructured region (UC), the CC plus UC (CC-UC), the five ankyrin repeats (ANKs), the UC plus ANK repeats (UC-ANKs) and the deletion mutant without the UC region (ΔUC) (Fig. 5a). The full-length NOS1APa and NOS1APc isoform as well as NOS1APc truncation mutants were N-terminally tagged with GFP. The latter included the PTB domain (PTB), exon 7-9 (E7-9), and the two exons encoding the C1orf226 (C1orf) (Fig. 5d). The immunoprecipitation experiments revealed that the UC-ANKs of KANK1 interacted weakly with NOS1APa and strongly with NOS1APc, the UC region interacted weakly with NOS1APc only and the remaining KANK1 mutants and deletion constructs failed to bind either of the NOS1AP isoforms (Fig. 5a–c). Conversely, full length NOS1APc, PTB, and to a lesser extent NOS1APa and C1orf region precipitated and hence, bound KANK1 (Fig. 5d–f).

Next, we generated home-made anti-peptide antibodies (Supplementary Fig. 7a, b) that recognize pan-NOS1AP, NOS1APa and NOS1APc to investigate where and how KANK1 colocalizes with NOS1AP in MCF7 cells and tissue sections. Consistent with the qRT-PCR result (Supplementary Fig. 6d), the non-canonical NOS1APc was readily detected by the pan-NOS1AP and NOS1APc-specific antibodies (Supplementary Fig. 7b), while the canonical NOS1APa isoform was barely detectable in

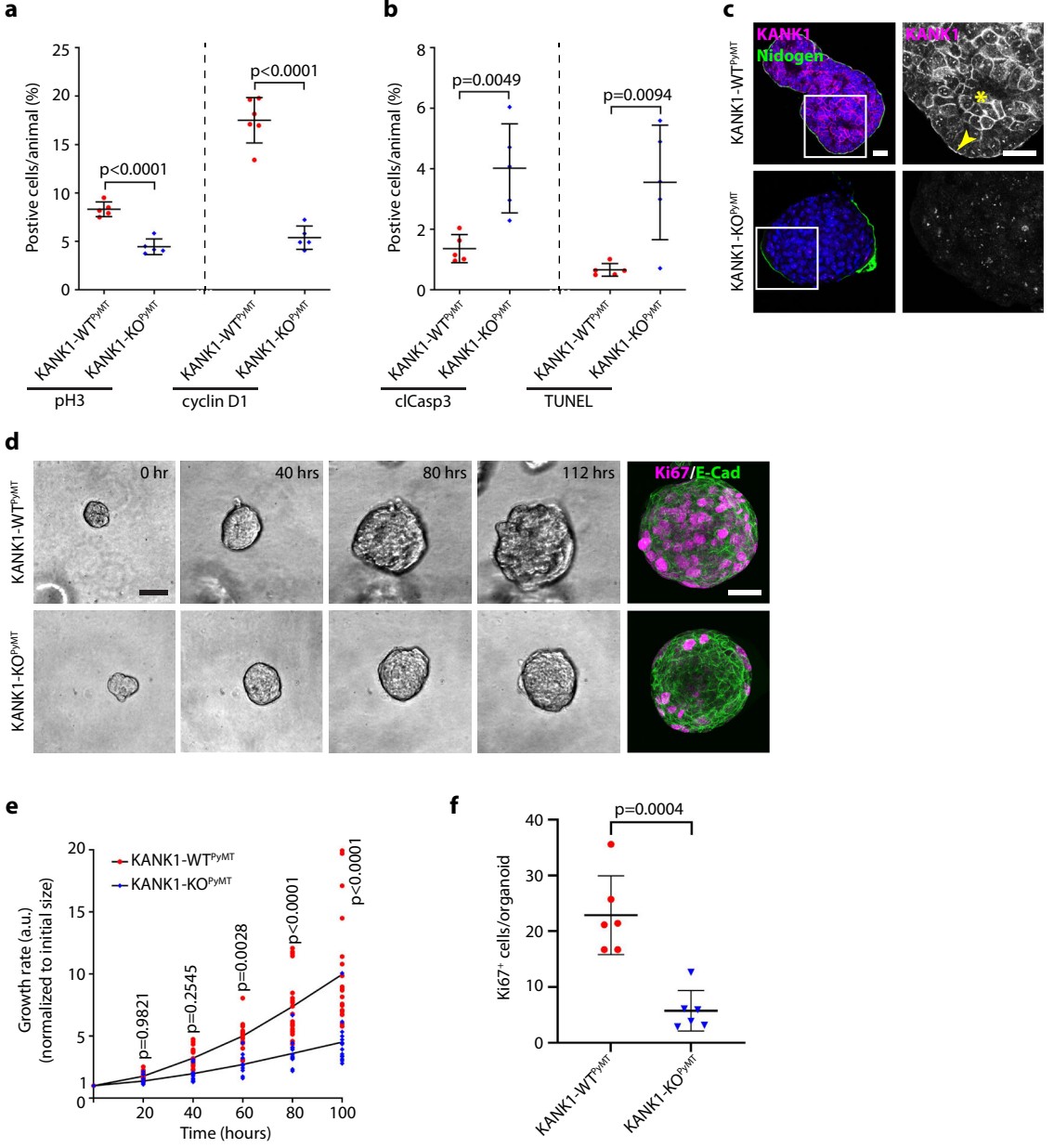

**Fig. 3 | KANK1 promotes tumor cell proliferation and survival. a, b** Quantification of cell proliferation analyzed by pH3 ($n = 5$ for each group) and cyclin D1 ($n = 6$ for KANK1-WT$^{PyMT}$ and 5 for KANK1-KO$^{PyMT}$) staining (**a**, see also Supplementary Fig. 5a), and cell survival analyzed by cleaved Caspase3 (clCasp3) staining and TUNEL assay (**b**, see also Supplementary Fig. 5b) ($n = 5$ for each group). *P* values (unpaired Student's *t* test, two-tailed) are indicated on top of each comparison. Mean ± SD is shown. **c** KANK1-WT$^{PyMT}$ and KANK1-KO$^{PyMT}$ tumoroids stained with KANK1 (magenta) and Nidogen (green). KANK1 localizes at basal side (arrow head) and cell-cell junctions (asterisk) of tumor cells. Nuclei were counterstained with DAPI (blue). Scale bars: 20 μm. **d** Stills of time-lapse imaging of tumoroid growth for 5 days.

Endpoint tumoroids were stained for Ki67 (magenta) and E-cadherin (E-cad, green). Maximal projection of the signals is shown. Scale bars: 50 μm. **e** Quantifications of KANK1-WT$^{PyMT}$ and KANK1-KO$^{PyMT}$ tumoroid growth rate ($n = 20$ for KANK1-WT$^{PyMT}$ and 15 for KANK1-KO$^{PyMT}$). Organoid size at the starting point was set as 1 (a.u.: arbitrary unit). *P* values were calculated by two-way ANOVA with Šídák's multiple comparisons test and indicated for each time point. All data points and mean are shown. **f** Quantifications of Ki67 positive (Ki67$^+$) cells per organoid ($n = 6$ for each group). *P* values were calculated by unpaired Student's *t*-test (two-tailed). Mean ± SD is shown. Source Data are provided as a Source Data file.

MCF7 cells. In line with these observations, the anti-NOS1APc isoform-specific antiserum showed strong expression at cell-cell junctions of CK8$^+$ LECs in normal mammary glands, KANK1-WT$^{PyMT}$, and KANK1-KO$^{PyMT}$ tumors, however, no expression in SMA$^+$ myoepithelial cells and cells of the tumor stroma (Fig. 6a). Anti-NOS1APa isoform-specific antiserum revealed strong expression in SMA$^+$ myoepithelial cells as well as smooth muscle cells that line blood vessels in normal mammary glands and KANK1-WT$^{PyMT}$ tumors (Supplementary Fig. 7c).

NOS1APc was shown to localize to the plasma membrane[36] suggesting that KANK1 binding to NOS1APc guides KANK1 to pan-

cadherin$^+$ cell-cell junctions in KANK1-WT$^{PyMT}$ tumors. To test this hypothesis, we retrovirally expressed mCherry-tagged NOS1APc or mCherry-only and a doxycycline (DOX)-inducible KANK1-GFP expression construct in NOS1AP Crispr/Cas9 knockout MCF7 cells (MCF7$^{NOS1AP-KO}$). Induced expression of KANK1-GFP in mCherry-NOS1APc-expressing cells showed KANK1-GFP in cell-cell junctions and paxillin-positive FAs (Fig. 6b1, b2, Supplementary Fig. 8a), whereas induced expression of KANK1-GFP in cells either not transduced with the NOS1APc retroviral construct (Fig. 6b1, asterisks) or transduced with mCherry-only (Fig. 6b3, b4) showed KANK1-GFP in FAs and

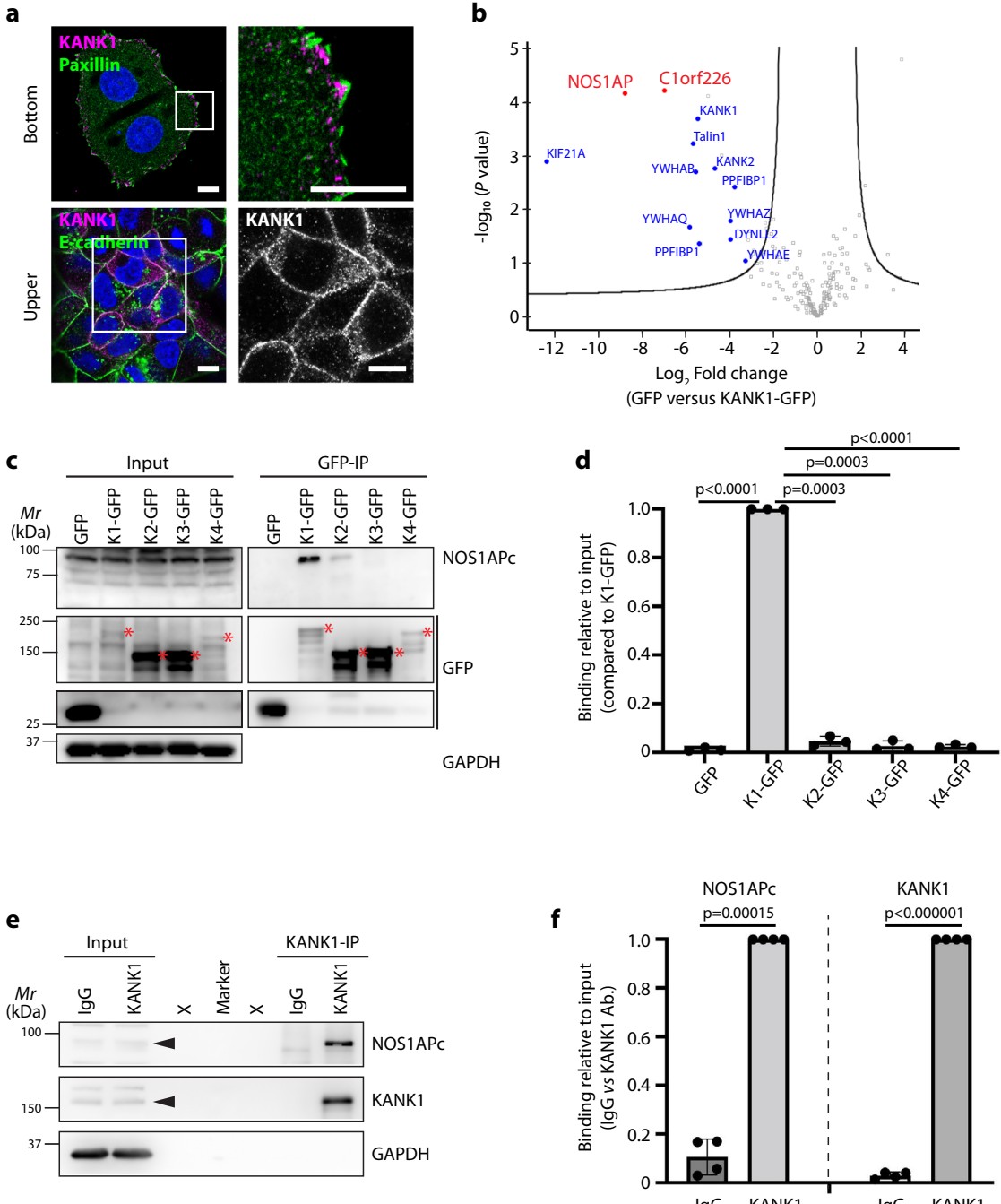

**Fig. 4 | NOS1APc is an interacting partner for KANK1. a** MCF7 cells stained for KANK1 (magenta), Paxillin (green) and E-cadherin (green). Nuclei were counterstained with DAPI (blue). $N = 4$ independent repeats. Scale bars: 10 μm. **b** Volcano plot of KANK1-GFP interacting proteins in MCF7 cells ($n = 3$ biological replicates). Known interactors are highlighted in blue, NOS1AP together with its splice isoform C1orf226 are highlighted in red. $P$ values were calculated by two-sided permuted $t$-test with 250 randomizations. The black line indicates the significance cut-off (FDR:0.08, s0:2) estimated by the Perseus software. **c, d** GFP immunoprecipitates (GFP-IP) in lysates from MCF7 cells expressing either GFP alone or GFP-tagged KANK1-KANK4 (K1-GFP to K4-GFP) blotted for pan-NOS1AP and GFP. Representative blot from 3 independent experiments. Expected bands for each KANK-GFP are indicated with red asterisks. GAPDH served as loading control (**c**). Quantification (**d**) was performed by normalizing the levels of NOS1APc after IP-enrichment to the corresponding input levels and the loading control. $P$ values were calculated by RM one-way ANOVA with multiple comparisons. Mean ± SD is shown. **e, f** Endogenous KANK1 immunoprecipitates (IP) in lysates from MCF7 cells blotted for pan-NOS1AP and KANK1. Representative blot from 4 independent experiments. Corresponding bands from input side are indicated with arrowheads. GAPDH served as loading control (**e**). Quantification (**f**) was performed by normalizing the levels of NOS1APc and KANK1 after IP-enrichment to their corresponding input levels and the loading control. $P$ values were calculated by multiple paired student $t$-tests (two-tailed). Mean ± SD is shown. Source Data are provided as a Source Data file.

diffusely distributed in the cytoplasm, but not in cell-cell junctions. Induced expression of KANK1-ΔUC-GFP localized to FAs, was diffusely distributed in the cytoplasm and, as expected from our mapping studies (Fig. 5a–c), absent from cell-cell junction of mCherry-NOS1APc-expressing cells (Fig. 6b5, b6). NOS1APc, on the other hand, localized

to cell-cell junctions and was absent from FAs, irrespective whether KANK1-GFP was expressed or not (Fig. 6b7, Supplementary Fig. 8a). Importantly, also the endogenously expressed KANK1 and NOS1APc colocalized at cell-cell junctions (Supplementary Fig. 8b). The absence of the KANK1-interacting FA proteins Talin-1 and Talin-2 as well as the

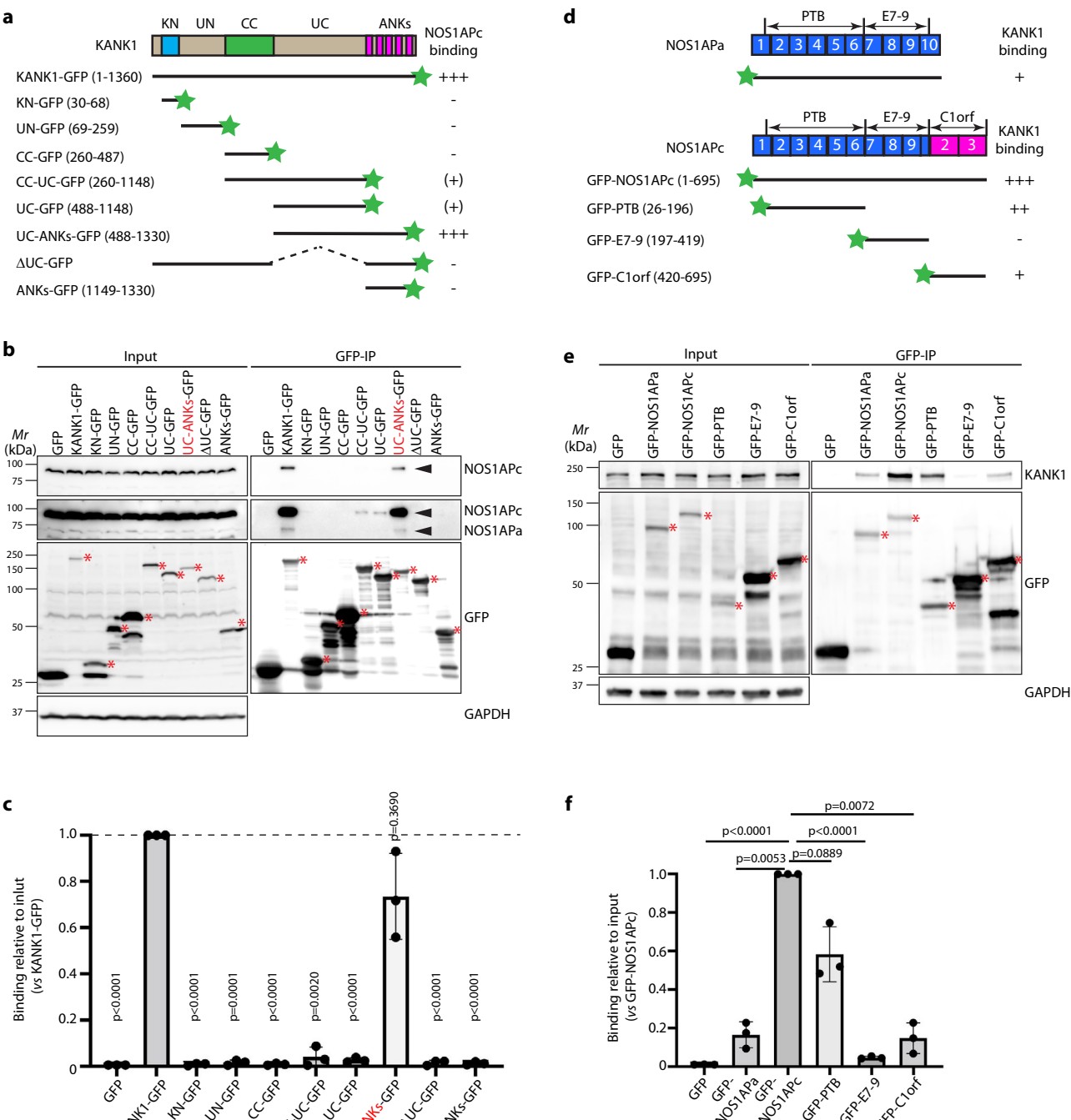

**Fig. 5 | NOS1APc binds to KANK1 c-terminal UC-ANKs region. a–c** Mapping of KANK1 domain binding to NOAS1APc. **a** Domain organization of KANK1 and illustration of C-terminally tagged GFP-KANK1 fragments. Strength of NOS1AP binding is shown on the right. +++ strong binding; - no binding; ( + ) indicates binding upon overexpression. **b** Representative blot from 3 independent experiments. The strongest interacting region is highlighted in red. Pan-NOS1AP antibody was used. Asterisks (red) indicate expected bands of GFP-tagged polypeptides. GAPDH served as loading control. **c** Quantification was performed by normalizing the levels of NOS1APc after IP-enrichment to their corresponding input levels and the loading control. *P* values were calculated by RM one-way ANOVA with Dunnett's multiple

comparisons. Mean ± SD is shown. **d–f** Mapping of NOAS1APc domain binding to KANK1. **d** Exon and domain organization of NOS1APc and illustration of N-terminally tagged GFP-NOS1APc fragments. Strength of KANK1 binding is shown on the right. **e** Representative blot from 3 independent experiments. Asterisks (red) indicate expected bands of GFP-tagged polypeptides. GAPDH served as loading control. **f** Quantification was performed by normalizing the levels of KANK1 after IP-enrichment to their corresponding input levels and the loading control. *P* values were calculated by RM one-way ANOVA with Dunnett's multiple comparisons. Mean ± SD is shown. Source Data are provided as a Source Data file.

CMSC protein KIF21A from cell-cell contacts (Supplementary Fig. 8c, d) suggests that the KANK1 interaction (1) with NOS1APc recruits KANK1 to cell-cell junctions, (2) with Talin to FAs[4,5], and (3) with KIF21A to the CMSC[6].

## KANK1 curbs SCRIB-NOS1AP mediated activation of Hippo signaling

NOS1AP is ubiquitously expressed and was shown to directly bind to the tumor suppressor and polarity protein SCRIB[44] (confirmed in

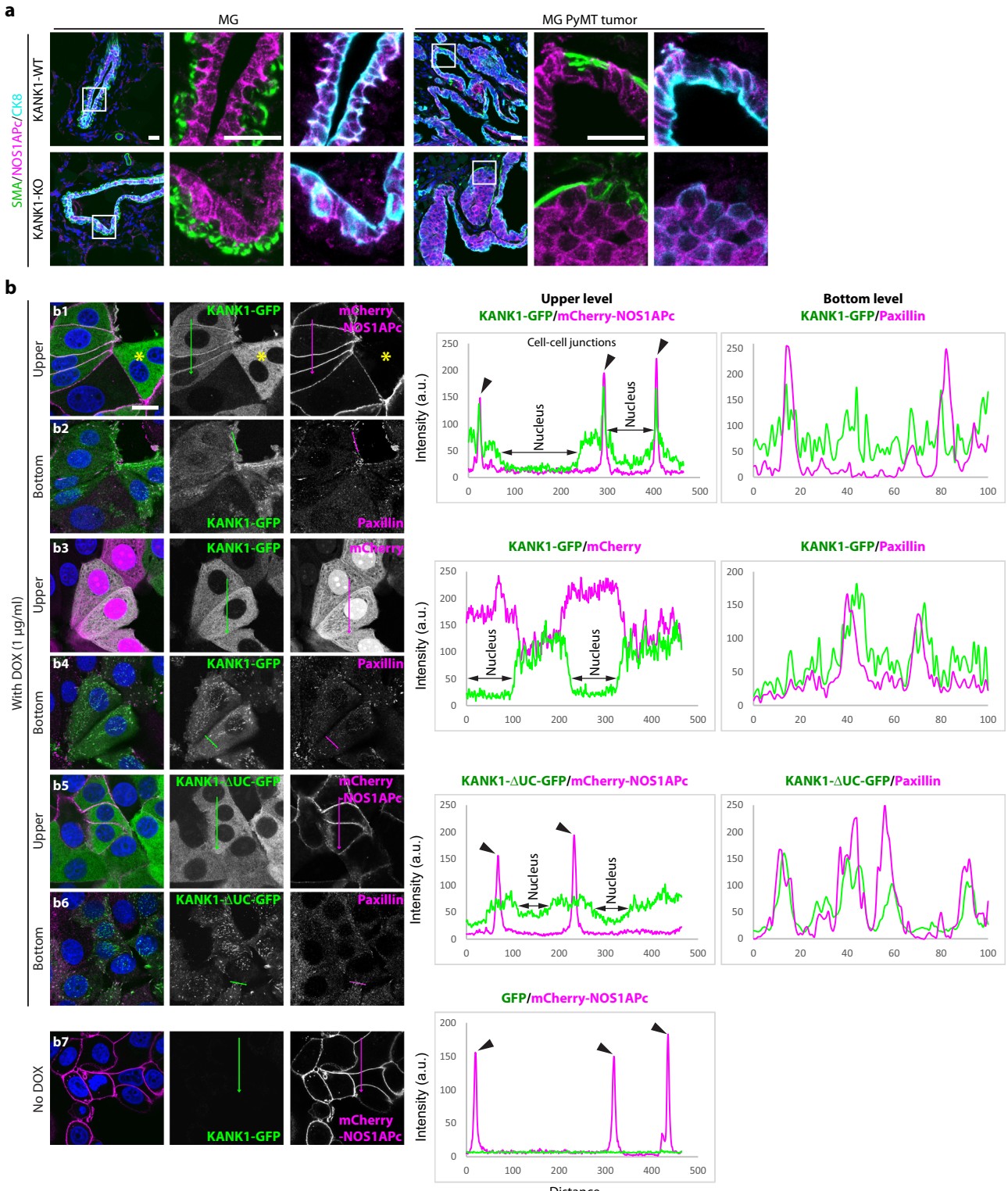

**Fig. 6 | NOS1APc recruits KANK1 to cell-cell junctions. a** Tissue sections of KANK1-WT and KANK1-KO mammary glands (MG) and KANK1-WT[PyMT] and KANK1-KO[PyMT] tumors stained for SMA (green), NOS1APc-specific antibody (magenta) and CK8 (Cyan). Nuclei were counterstained with DAPI (blue). $N = 4$ independent repeats. Scale bars: 20 μm. **b** MCF7[NOS1AP-KO] cells were virally reconstituted with mCherry-NOS1APc (b1-2, b5-7) or mCherry only (b3-4) together with DOX-inducible KANK1-GFP expression construct (b1-4) or with DOX-inducible KANK1-ΔUC-GFP (b5-6) and stained for Paxillin to highlight FAs at the bottom layer. The two cells not transduced with the mCherry-NOS1APc expression construct (asterisks, b1) displayed KANK1-GFP in the cytoplasm but not cell-cell contact. Black arrow heads in the line profile analysis (right panel) indicate the co-localization of KANK1-GFP and mCherry-NOS1APc at cell-cell junctions ($n = 5$ independent repeats). Scale bar: 20 μm. Source Data are provided as a Source Data file.

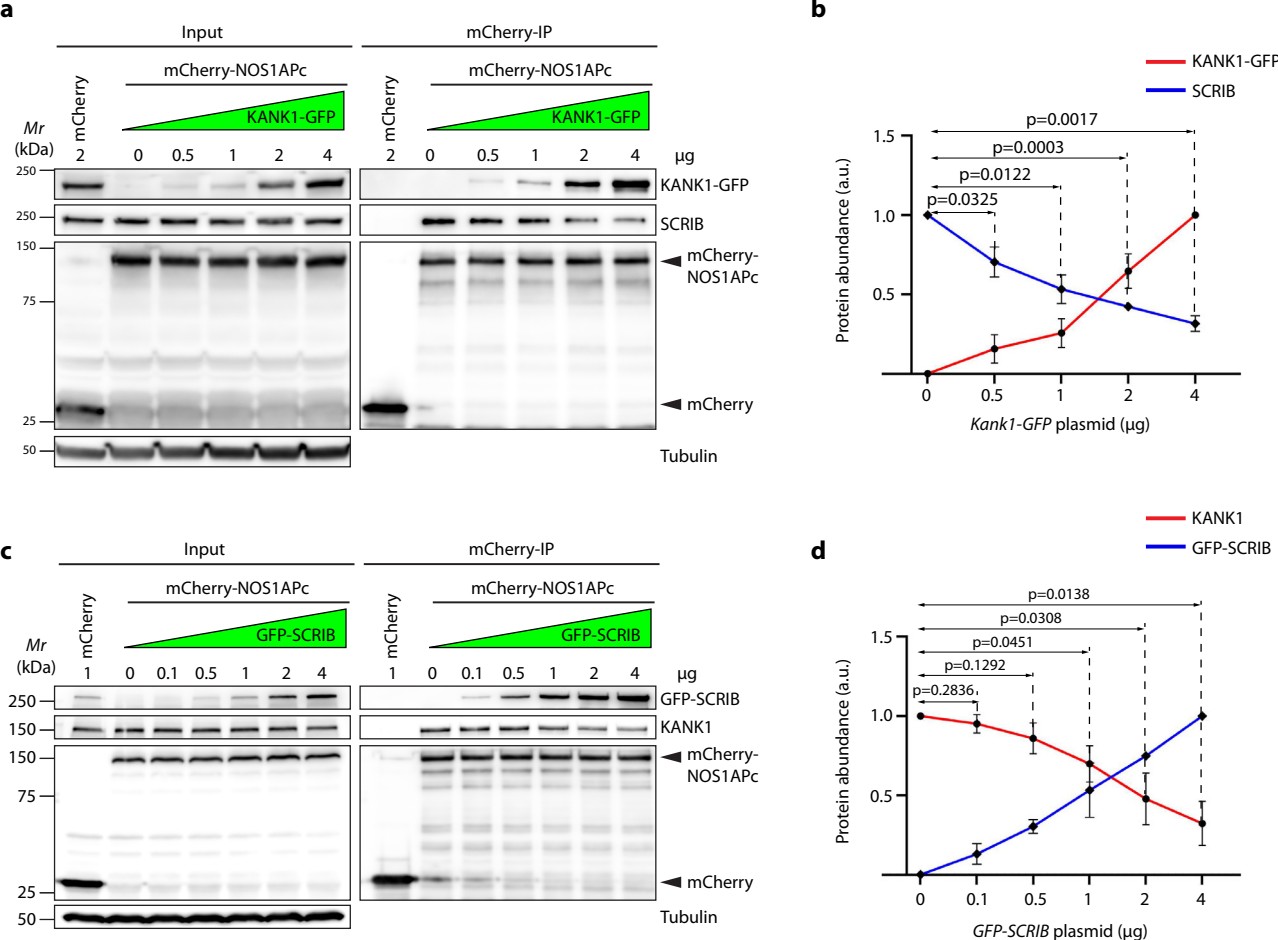

**Fig. 7 | KANK1 competes with SCRIB for NOS1APc binding. a, b** WB analysis of mCherry-NOS1APc immunoprecipitates (IP) for KANK1-GFP, SCRIB, mCherry-NOS1APc in HEK293T cells transfected with increasing levels of KANK1-GFP cDNA. mCherry cDNA transfection served as control. Tubulin served as loading control. Representative blot from 3 independent experiments (**a**). Protein abundance of KANK1-GFP and SCRIB normalized to input Tubulin levels. Value of SCRIB level without exogenous KANK1-GFP competition was set as 1 (a.u.: arbitrary unit). *P* values were calculated by one sample *t*-test (two-tailed). Mean ± SD is shown (**b**).

**c, d** WB analysis of mCherry-NOS1APc immunoprecipitates (IP) for GFP-SCRIB, KANK1, mCherry-NOS1APc in HEK293T cells transfected with increasing levels of GFP-SCRIB cDNA. mCherry cDNA transfection served as control. Tubulin served as loading control. Representative blot from 3 independent experiments (**c**). Protein abundance of KANK1 and GFP-SCRIB normalized to input Tubulin levels. Value of KANK1 level without exogenous GFP-SCRIB competition was set as 1 (a.u.: arbitrary unit). *P* values were calculated by one sample *t*-test (two-tailed). Mean ± SD is shown (**d**). Source Data are provided as a Source Data file.

Supplementary Fig. 9a). Since both KANK1 and SCRIB[44] bind the PTB domain of NOS1APc, we tested whether KANK1 and SCRIB compete for NOS1AP binding by performing competitive binding assays in HEK293T cells. We transduced increasing concentrations of the KANK1-GFP expression vector in mCherry-tagged NOS1APc expressing HEK293T cells and found that the complexes between endogenous SCRIB and mCherry-NOS1APc decreased with increasing KANK1 levels (Fig. 7a, b). Conversely, the transduction of increasing concentrations of GFP-SCRIB expression vectors decreased complexes between endogenous KANK1 and mCherry-NOS1APc (Fig. 7c, d).

The integrity of the SCRIB protein and the membrane localization of SCRIB is essential to bind and activate the kinase cascade MST1/2 and LATS1/2 of the Hippo pathway, which leads to YAP and TAZ phosphorylation, TAZ instability and low nuclear YAP/TAZ levels in mammary epithelial cells[37,38]. Conversely, disabling SCRIB, e.g. by dislocation from the plasma membrane into the cytoplasm, impairs the SCRIB influence on the Hippo pathway and leads to increased TAZ stability, nuclear localization of TAZ, increased breast cancer cell proliferation and breast cancer growth[37,41]. Since not only SCRIB but also NOS1AP can activate the Hippo pathway and regulate YAP/TAZ levels in mammary epithelial cells[36,37], we hypothesized that the

competitive removal of NOS1APc from SCRIB by KANK1 compromises SCRIB, akin to the plasma membrane dislocation, and thereby impairs SCRIB-mediated Hippo signaling. To test this hypothesis, we deleted the *KANK1* alleles in MCF7 cells with Crispr/Cas9 (MCF7[KANK1-KO]) and investigated the influence of KANK1 loss on integrin and YAP/TAZ protein levels. The integrin surface levels determined by flow cytometry were unaffected in MCF7[KANK1-KO] cells (Supplementary Fig. 9b). The TAZ protein levels but not the *TAZ* mRNA levels (Fig. 8a, b, Supplementary Fig. 9c) were significantly higher in MCF7[KANK1-WT] compared to MCF7[KANK1-KO] cells, suggesting that KANK1 regulates TAZ protein stability. The YAP protein levels remained unaffected upon loss of KANK1 (Fig. 8a). Treatment of KANK1-deficient MCF7[KANK1-KO] cells with cycloheximide (CHX), which blocks protein synthesis, revealed that the TAZ protein remained unaffected in CHX-treated MCF7[KANK1-WT] cells, whereas the TAZ protein decreased by 40% within 6 h in MCF7[KANK1-KO] cells (Fig. 8c, d). Since the TAZ protein levels in MCF7[KANK1-KO] cells were stabilized by the proteasome inhibitor MG132 but not the lysosome inhibitor Bafilomycin A1 (BAF) (Fig. 8a, b), we conclude that in the absence of KANK1, TAZ is degraded by the UPS.

The phosphorylation of YAP/TAZ at specific residues by the activity of the MST/LATS kinase cascade dictates whether YAP/TAZ is

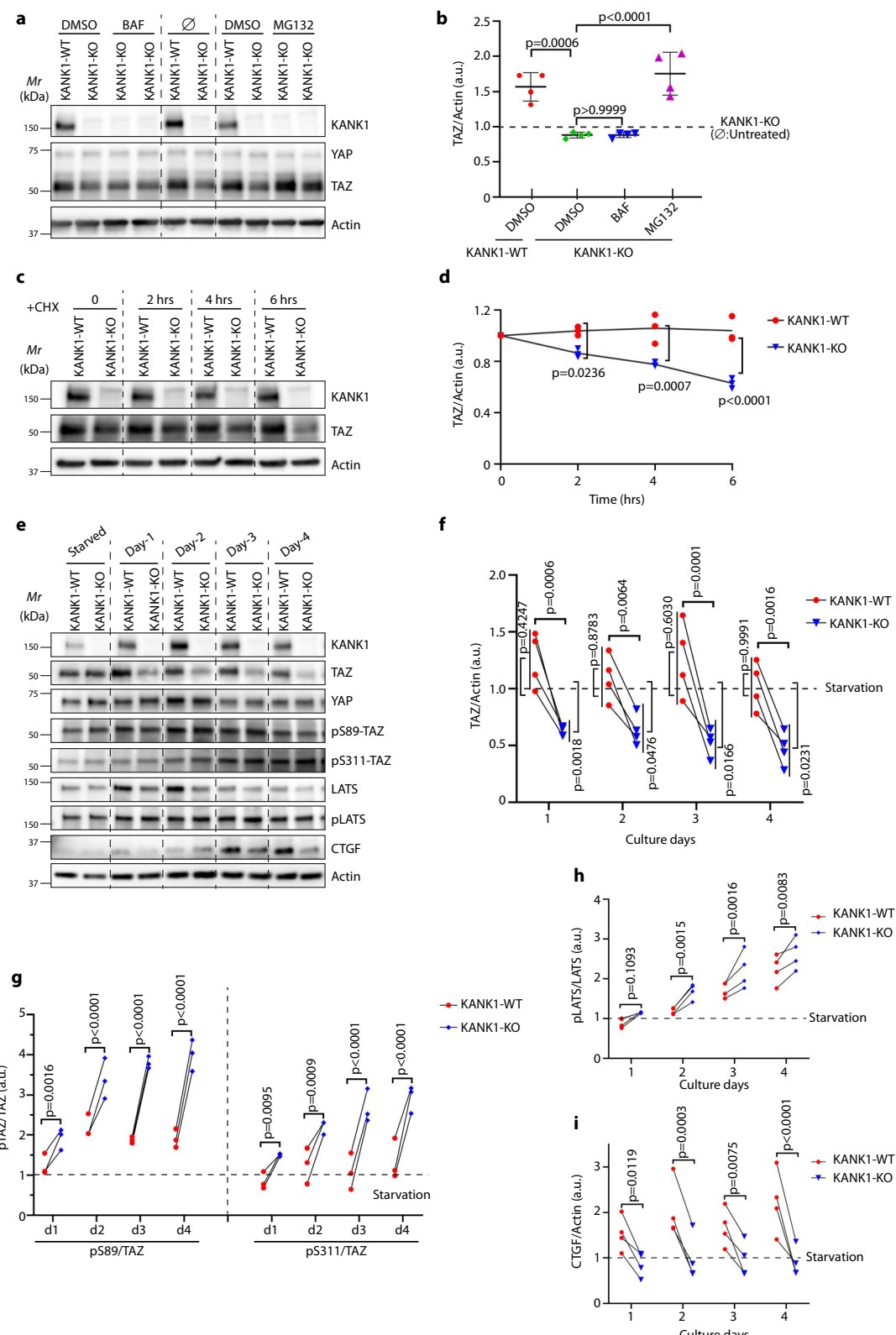

degraded by UPS or bound by 14-3-3 proteins and retained in the cytoplasm[32,33]. In the case of TAZ, phosphorylation of serine-89 (pS89-TAZ) leads to 14-3-3 binding and cytoplasmic retention, and phosphorylation of serine-311 (pS311-TAZ) to proteasomal degradation[33]. To determine the phosphorylation state of TAZ, we treated serum-starved MCF7[KANK1-WT] and MCF7[KANK1-KO] cells with serum for 4 days and immunoblotted cell lysates with specific antibodies. The total, pS89- and pS311-

TAZ levels were comparable between serum-starved MCF7[KANK1-WT] and MCF7[KANK1-KO] cells. However, addition of serum considerably decreased TAZ levels in MCF7[KANK1-KO] cells, while the TAZ levels remained unchanged in MCF7[KANK1-WT] cells (Fig. 8e, f). Normalization of pS89- and pS311-TAZ levels to total TAZ levels revealed a significant reduction of pS89- as well as pS311-TAZ in serum-treated MCF7[KANK1-WT] compared to MCF7[KANK1-KO] cells (Fig. 8e, g), which was consistent with the

**Fig. 8 | KANK1 stabilizes the TAZ protein by preventing proteasomal degradation. a, b** WB analysis of YAP, TAZ in lysates of MCF7[KANK1-WT] and MCF7[KANK1-KO] cells treated with either DMSO, Bafilomycin A1 (BAF) or MG132 (n = 4 biological replicates). Actin served as loading control. Relative TAZ protein level (normalized to Actin) is shown in (**b**). Value of TAZ level in untreated (∅) KANK1-KO cells was set as 1 (a.u.: arbitrary unit). *P* values were calculated using one-way ANOVA with Dunnett's multiple comparisons test and are indicated on top of each comparison. Mean ± SD is shown. **c, d** Protein stability analyzed upon Cycloheximide (CHX) treatment for times indicated (**c**) (n = 3 biological replicates). Relative TAZ protein level (normalized to Actin) is shown in (**d**). Value of TAZ level at starting point was set as 1 (a.u.: arbitrary unit). *P* values were calculated using two-way ANOVA with Šídák's multiple comparisons test and are indicated for each time point comparison. **e, f** WB analysis of indicated proteins in lysates of serum-treated MCF7[KANK1-WT]

and MCF7[KANK1-KO] cells after overnight serum starvation. Representative blot from 4 independent experiments. Actin served as loading control (**e**). Relative TAZ protein level (normalized to Actin) is shown in (**f**). TAZ level before serum treatment (Starvation) was set as 1 (a.u.: arbitrary unit). *P* values were calculated using RM two-way ANOVA with Šídák's multiple comparisons test and are indicated on top of comparison between KANK1-WT and KANK1-KO, and indicated at the lateral side of comparison between each condition with starvation level. **g–i** Densitometric analysis of pS89-TAZ/TAZ and pS311-TAZ/TAZ (**g**, n = 3), pLATS/LATS (**h**, n = 4) and CTGF/Actin protein levels (**i**, n = 4) in MCF7[KANK1-WT] and MCF7[KANK1-KO] lysates from the representative analysis shown in (**e**). Protein levels before serum treatment (Starvation) were set as 1 (a.u.: arbitrary unit). *P* values were calculated using RM two-way ANOVA with Šídák's multiple comparisons test. Source Data are provided as a Source Data file.

simultaneously higher activity of the upstream LATS kinase in MCF7[KANK1-KO] cells when normalizing to total LATS level (Fig. 8e, h). Concomitantly with the stabilization of TAZ in serum-treated MCF7[KANK1-WT] cells, the mRNA levels of the TAZ target genes coding for connective tissue growth factor (*CTGF*) and cysteine-rich heparin binding protein 61 (*CYR61*) (Supplementary Fig. 10a) as well as the CTGF protein levels were also elevated in MCF7[KANK1-WT] cells (Fig. 8e, i, Supplementary Fig. 9d). These increased phosphorylation and decreased stabilization of TAZ were independently confirmed in a second MCF7[KANK1-KO] cell clone (Supplementary Fig. 10a–c). The elevated TAZ protein levels in MCF7[KANK1-WT] cells were associated with increased nuclear levels of TAZ (Supplementary Fig. 11a, b). Importantly, re-expression of KANK1-GFP in MCF7[KANK1-KO] cells re-established KANK1-GFP localization at cell-cell junctions, nuclear TAZ and increased total TAZ levels upon serum treatment (Supplementary Fig. 11c–f), whereas expression of the NOS1AP-binding deficient KANK1-ΔUC-GFP or GFP-only failed to rescue nuclear TAZ localization (Supplementary Fig. 11c, d).

## TAZ stabilization abolishes the defects caused by KANK1 loss

Our findings indicate that KANK1 localization at the cell-cell junction of LECs from multilayered carcinomas leads to NOS1AP binding, disassembly of the SCRIB/NOS1AP complex, which profoundly compromises SCRIB's ability to activate the Hippo pathway and stabilizes TAZ. If NOS1AP supports the tumor suppressor function of SCRIB, depletion of NOS1AP in MCF7[KANK1-KO] cells should foil the effect of KANK1 on SCRIB-mediated TAZ stability regulation. To test this hypothesis, we siRNA-depleted NOS1AP in MCF7[KANK1-KO] cells and observed reduced pS89- and pS311-TAZ levels, stabilization of total TAZ levels and normalization of CTGF protein levels 72hrs later (Supplementary Fig. 12a, b). Similarly, and as reported earlier by others[37], siRNA-mediated depletion of SCRIB in MCF7[KANK1-KO] cells also stabilized total TAZ levels (Supplementary Fig. 12c, d).

Next, we retrovirally transduced MCF7[KANK1-KO] cells with the DOX-inducible HA-tagged wildtype (HA-tagged MCF7[KANK1-KO]-TAZ[WT]; abbreviated with TAZ[WT]) or mutant TAZ in which the LATS-targeted serine-89 and serine-311 phosphorylation sites were substituted for non-phosphorylatable alanine residues (HA-tagged MCF7[KANK1-KO]-TAZ[S89A/S311A]; abbreviated with TAZ[SSAA]), which will stabilize the TAZ protein[32,33]. Expectedly, the *CTGF* as well as *CYR61* mRNA levels were lower in MCF7[KANK1-KO] cells compared to MCF7[KANK1-WT] cells at Day-1 and Day-3 of analyses. Upon overexpression of TAZ[WT], the *CTGF* and *CYR61* mRNA levels increased at Day-1 in MCF7[KANK1-KO] cells and reached levels of untreated MCF7[KANK1-WT] cells, whereas at Day-3 *CTGF* and *CYR61* mRNA levels decreased to untreated MCF7[KANK1-KO] cells, suggesting the overexpressed TAZ[WT] protein is degraded in the absence of KANK1 resulting in a decrease of the *CTGF* and *CYR61* transcription. However, in TAZ[SSAA]-expressing MCF7[KANK1-KO] cells the *CTGF* as well as *CYR61* mRNA levels increased at Day-1 and further at Day-3 indicating that stabilizing TAZ overrides KANK1 loss (Supplementary Fig. 13a). In

line with this hypothesis, expression of the stabilized TAZ[SSAA] in MCF7[KANK1-KO] tumoroids grown over a 5-day period in Matrigel reversed the growth deficit of non-transduced MCF7[KANK1-KO] cells (Supplementary Fig. 13b, c, Supplementary Movie 3–5). It will be important to demonstrate in future experiments that injection of TAZ[SSAA] expressing MCF7[KANK1-KO] cells into the fat pad of immunodeficient mice is overriding the diminished tumor growth also in vivo.

Since TAZ can endow self-renewal capacity to breast cancer cells[37], we tested whether KANK1 modulates breast cancer cell renewal and whether stabilized TAZ[SSAA] can overturn potential defects in MCF7[KANK1-KO] cells. To this end we measured mammosphere formation between MCF7[KANK1-WT], MCF7[KANK1-KO] and MCF7[KANK1-KO]-TAZ[SSAA] cells. The experiments revealed that MCF7[KANK1-KO] cells formed fewer and smaller mammospheres compared to MCF7[KANK1-WT] cells and that expression of the non-phosphorylatable TAZ[SSAA] normalized mammosphere numbers and size (Supplementary Fig. 14a–c).

## KANK1 expression correlates with nuclear TAZ in mouse and human breast cancer

If cell-cell junction localized KANK1 compromises SCRIB-mediated Hippo signaling by competing for NOS1AP binding, nuclear TAZ levels should be elevated in KANK1-WT[PyMT] compared to KANK1-KO[PyMT] tumors. Immunostaining of KANK1-WT[PyMT] tumor sections revealed that KANK1 localized at cell-cell junction and TAZ in the nucleus (Fig. 9a, asterisks), except the tumor cells that were still attached to the BM. They retained KANK1 at their basal side and TAZ predominantly in the cytoplasm (Fig. 9a, arrow heads). In contrast, KANK1-KO[PyMT] tumors showed TAZ primarily cytoplasmic, irrespective whether tumor cells were attached to or away from BM (Fig. 9a, b). Similarly, elevated nuclear TAZ levels with KANK1 at cell-cell junctions were also observed in KANK1-WT[PyMT] tumoroids (Fig. 9c–e), which explains their accelerated growth compared to KANK1-KO[PyMT] tumoroids (Fig. 3d–f).

Next, we performed a series of experiments to confirm the relevance of our findings from the KANK1-WT[PyMT] mouse breast cancer model for human breast cancer. First, we found that KANK1 expression was restricted to epithelial tumor cells and absent from the surrounding stroma in samples of luminal breast cancers (Supplementary Fig. 15a). Second, colocalization of KANK1, NOS1APc and cadherin were observed at cell-cell junctions and TAZ in the nucleus of human breast cancer cells (Fig. 10a). This finding was further supported by the positive correlation between the subcellular localization of KANK1 at plasma membrane and TAZ in the nucleus[56] (Fig. 10b), Third, inspection of the TCGA revealed a positive correlation between *KANK1* mRNA levels and mRNA levels of the TAZ targets, *CTGF* and *CYR61* in luminal breast cancers (Supplementary Fig. 15b–g). Finally, we injected human MCF7[KANK1-WT] and MCF7[KANK1-KO] cells into the fat pad of immunodeficient mice and compared their growth during an 8-week period. The experiment revealed that MCF7[KANK1-WT] tumors were visible already 2 weeks after implantation, whereas MCF7[KANK1-KO] tumors were first detected 5-6 weeks after implantation (Fig. 10c, d). The MCF7[KANK1-WT]

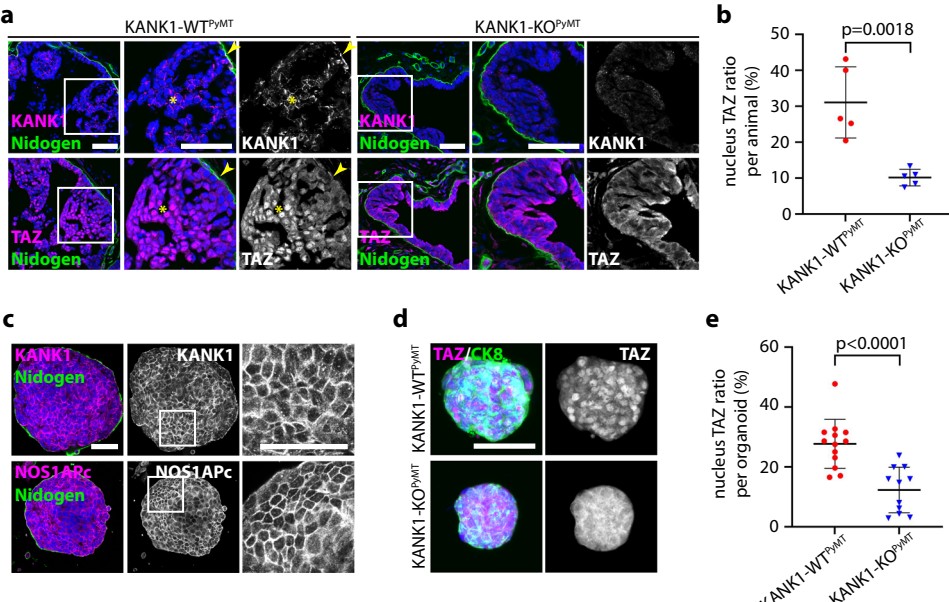

**Fig. 9 | Junctional translocation of KANK1 promotes nuclear TAZ accumulation in malignant tissues. a, b** Tumor sections from KANK1-WT[PyMT] and KANK1-KO[PyMT] mice stained for KANK1 or TAZ (magenta) and Nidogen (green). Arrow heads indicate BM-attached KANK1-WT[PyMT] tumor cells with predominantly cytoplasmic TAZ. Nuclei were counterstained with DAPI. Scale bars: 50 μm (**a**). The ratio of TAZ-positive nuclei per animal is shown in (**b**) ($n = 5$ for each group). $P$ value was calculated using unpaired Student's $t$-test (two-tailed). Mean ± SD is shown. **c** Tumoroids cultured in Matrigel were whole-mount stained for KANK1 (magenta) or NOS1APc-specific antibody (magenta) and Nidogen (green). Nuclei were counterstained with DAPI (blue). Repeats: $n = 7$ tumoroids for KANK1 and $n = 8$ for NOS1APc. Scale bars: 50 μm. **d, e** Tumoroids were whole-mount stained for TAZ (magenta) and CK8 (green). Maximal projection of the signals is shown. Nuclei were counterstained with DAPI. Scale bar: 50 μm (**d**). The ratio of TAZ-positive nuclei per organoid is shown in (**e**). Repeats: $n = 13$ for KANK1-WT[PyMT] and $n = 11$ for KANK1-KO[PyMT]. $P$ value was calculated using unpaired Student's $t$-test (two-tailed). Mean ± SD is shown. Source Data are provided as a Source Data file.

tumors also showed faster growth rate and were larger with more nuclear TAZ at the endpoint of the experiment when compared with MCF7[KANK1-KO] tumors (Fig. 10d, Supplementary Fig. 15h–k). These data demonstrate that the tumor promoting function of KANK1 is shared between mouse and human.

## Discussion

The KANKs are FA proteins that have so far been shown to link integrin adhesions to the adjacent protein complex (cortical microtubule stabilizing complex; CMSC) that organizes and stabilizes cortical microtubules[2,3,9]. Since it is not entirely clear whether KANK1 promotes or curbs tumor development in vivo, we decided to analyze breast cancer growth in mice and tumoroids lacking KANK1 expression.

KANK1-deficient mice developed and aged normally and displayed no mammary gland defects despite the high de novo KANK1 expression during pregnancy and lactation. When we induced breast cancer with the PyMT transgene, we found that KANK1-WT[PyMT] suffered from a significantly higher tumor incidence and tumor load than their KANK1-KO[PyMT] littermates. The accelerated tumor development in KANK1-WT[PyMT] mice was due to an elevated tumor cell proliferation and survival. In search for a mechanistic explanation for the increased tumor growth in KANK1-WT[PyMT] mice, we discovered several steps of an oncogenic signaling pathway, in which KANK1 undertakes the principal task to impair the tumor suppressive function of the cell polarity protein SCRIB. In a first step loss of cell polarity, which occurs when cancer cells lose the BM contacts, grow into the ductal lumen and form multilayered stacks. At this tumor stage, KANK1 expression switches from a predominant localization at integrin adhesion sites to cell-cell junctions. Following this switch, KANK1 competes with SCRIB for NOS1AP binding, which in turn compromises SCRIB-mediated Hippo signaling leading to significantly decreased phosphorylation of TAZ, increased stabilization of TAZ and elevated nuclear accumulation of TAZ followed by promoting the tumor-initiating potential of breast

cancer stem cells as well as the transcription of growth promoting and cell survival genes (Fig. 10e). Importantly, the same KANK1-regulated TAZ signaling described here for transformed luminal epithelial cells (LECs) in mice is also operating in the human breast cancer cell line MCF7 transplanted into immunocompromised mice and in human breast cancer. Immunostaining of tumor sections from human breast cancer patients revealed that the majority of tumor cells contained KANK1 at cell-cell junctions and TAZ in the nucleus, and those LECs still attached to the BMs contained KANK1 at their basal, BM-binding side and TAZ diffusely in the cytoplasm.

Our findings show that the recruitment of KANK1 to cell-cell junctions in breast cancer cells of KANK1-WT[PyMT] mice is a stepwise process. In BM-bound LECs, KANK1 is found at basal integrin adhesion sites and is excluded from cell-cell junctions. Once transformed, LECs lose their contact with the BM leading to the disassembly of integrin adhesions and the release of KANK1 and Talin into the cytoplasm, where Talin adopts an inactive conformation unable to further associate with KANK1[4,5,57,58]. The unleashed KANK1 is able to engage the PTB domain of NOS1AP and freely move to the plasma membrane and thereby compromises the SCRIB/NOS1AP complex resulting in the inactivation of the Hippo pathway, which ensures that TAZ levels remain low and cytoplasmic in normal LECs. The competition for NOS1AP binding by SCRIB and KANK1 indicates that KANK1 can only remain at the cell-cell junction and impair SCRIB if it 'wins' the binding competition, probably by a superior binding affinity. In light of this complex molecular choreography, and its important role for breast cancer, it will be important to biochemically and structurally determine the binding characteristics of SCRIB, NOS1AP and KANK1 in future.

Finally, we demonstrate that KANK1 is exclusively expressed in transformed LECs, while KANK2-4 are expressed in stromal cells, which strongly indicates that the oncogenic KANK1-mediated Hippo signaling operates in a tumor cell-autonomous manner and is not

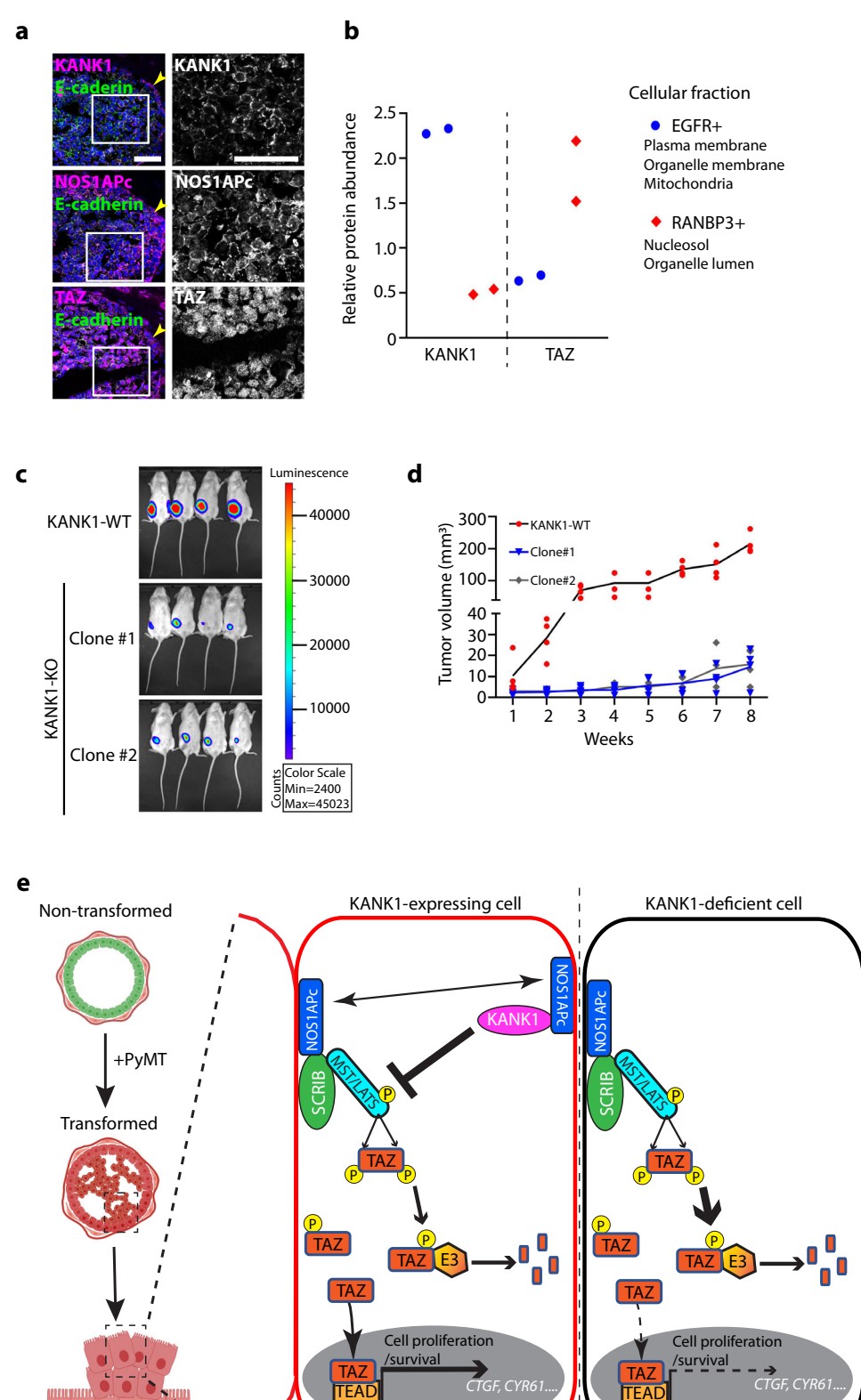

influenced by stromal cell types that do not express KANK1. The tumoroids established from PyMT transformed LECs confirm that the association of KANK1 with NOS1AP at cell-cell junctions suffice to induce nuclear TAZ accumulation and enhance cell proliferation. It is conceivable, however, that tumor cells that break through the BM and invade the stiff tumor stroma, reinforce the activation of TAZ, and probably also YAP by mechanotransduction[29,59]. In this situation, the displacement of KANK1 from FAs and CMSCs may cooperate with the biomechanical signals from the tumor environment via destabilizing microtubules and activating RhoA leading to stress fiber formation and myosin II activation, and enhancing the actomyosin-Talin linkage and force transmission across integrins leading to adhesion strengthening and enhanced tumor cell migration and invasion.

**Fig. 10 | KANK1 in human breast cancer. a** Human breast cancer tissues immunostained for KANK1 (magenta), NOS1APc-specific antibody (magenta), TAZ (magenta) and E-cadherin (green). Arrow heads indicate tumor cells at the basal side with predominantly cytoplasmic TAZ. Nuclei were counterstained with DAPI. Tissues were from 3 independent breast cancer patients. Scale bars: 50 μm. **b** Relative protein abundance of KANK1 and TAZ in subcellular fractions of MCF7 cell lysates. Protein abundance in EGFR[+] plasma membrane fraction and RANBP3[+] nucleosol fraction is shown ($n = 2$ biological replicates). Note that high KANK1 expression at the plasma membrane correlates with higher TAZ in the nucleus.

**c** MCF7[KANK1-WT] and MCF7[KANK1-KO] (2 independent clones) cells were luciferase-labeled and then orthotopically injected into the fat pad of immunodeficient mice. Luciferase intensity indicated by color grading in tumors at the experimental end point are shown. **d** Weekly monitoring of tumor volume after fat pad injection ($n = 4$ mice analyzed for each group). All data points and mean are shown. **e** Model illustrating the proposed mechanism of Hippo signaling pathway by the interplay of KANK1, NOS1APc and SCRIB (Created in BioRender. Guo, S. (2023) BioRender.com/x85q744). Double arrow heads (↔) indicate the competition between SCRIB and KANK1 for NOS1APc binding. Source Data are provided as a Source Data file.

The limitation of the current study is that the significance of the stabilized TAZ[SSAA] was not validated in a KANK1-KO background in vivo. Although we demonstrate that expression of TAZ[SSAA] in MCF7[KANK1-KO] tumoroids reversed the growth retardation of non-transduced MCF7[KANK1-KO] tumoroids, xenografting TAZ[SSAA]-expressing MCF7[KANK1-KO] cells in vivo will constitute strong evidence for KANK1 inhibition as a potential therapeutic strategy to curb breast cancer growth in patients.

## Methods

### Mouse strains

The conditional *Kank1*[fl/fl] mouse strain with loxp-flanked Exon 6 and 7 (Supplementary Fig. 1a) were generated via electroporation of R1 ES cells using standard procedures[60] and were backcrossed to C57BL/6 background for 10 generations. Homologous recombination was verified with southern blots and positive ES clones were used to generate germline chimeric mice. To achieve a constitutive deletion of *Kank1* gene, mice were crossed with transgenics carrying a deletor-Cre[48]. The MMTV-PyMT (stock NO. 022974) mouse strain on C57BL/6 genetic background was obtained from the Jackson Laboratory. The mice were housed in rooms at 22 °C with 55% of humidity, with light:dark cycle of 14:10 h and with autoclaved water in bottles.

Tumor growth was monitored by weekly palpation. Once tumor was palpated, the length and the width of the tumors were measured weekly with a caliper till the end of the experiments. Tumor volume was calculated as described before[61]. End points were set when a single tumor reached a diameter of ≥1,5 cm or several tumors reached a sum diameter of ≥3 cm. At experimental end points, mice were euthanized and tumors were dissected out for further analysis, such as photographing, weighing or fixation and staining. All mouse experiments were performed according to the regulations approved by the government of Upper Bavaria, Germany (license NO: 55.2-2532.Vet_02-20-182), and the Committee on the Use of Live Animals in Teaching and Research of the University of Hong Kong (CULATR NO: 5842-21).

### Hematoxylin-Eosin (HE) staining and tumor grading

Tumors were dissected out for overnight fixation in 4% PFA-PBS, followed by dehydration and tissue clearance in Xylene before paraffin-embedding. Paraffin blocks were then sectioned at a thickness of 5 μm for HE staining (Hematoxylin solution: Sigma, HHS32; Eosin Y: Sigma, HT1101). To obtain a good overview for tumor grading, a total of 120–150 sections were collected from each tumor block. HE staining were performed in sections at an interval of 10 sections. Hence, a total of 10–15 sections from each tumor mice were used for histological grading.

### Immunofluorescence microscopy

For immunostaining, cells were cultured on glass coated with 10 μg/ml FN (Calbiochem®, #341631) at 37 °C for 1 h, then fixed with cold methanol:acetone (1:1) for 10 min at −20 °C (for Talin staining) followed by 15 min rehydration in PBS at RT, or with 4% PFA − PBS for 10 min at RT (for all other immunostaining). PFA-fixed cells were permeabilized with 0.1% Triton X-100−PBS for 15 min at RT. Fixed cells were blocked with 3% BSA in PBS for 2 hours at RT followed by

incubation with the primary antibodies in 3% BSA in PBS overnight at 4 °C and then with secondary antibodies for 2 h at RT.

Virgin MGs were isolated from 4-month-old female mice, MGs from female mice at puberty were harvested from 5-week-old offspring, from pregnant mice at mid-pregnancy (14 days after mating), from lactating mice 10 days after delivery, and from mice one month after weaning (involuting MGs). Tumor tissues were obtained from 16-week-old littermates unless indicated otherwise. The isolated tissues were fixed at 4 °C for 5 h before they were embedded in Cryomatrix™ (Epredia). Tissue sections were prepared by Cryostar NX70 (Epredia) at a thickness of 7 μm. Snap-frozen clinical human breast cancer tissues were embedded and sectioned first, and then fixed for 15 min at RT before staining. Written consent for usage of patient tumor tissues for research purpose was provided by the patients or their family members, which was approved by the Ethics Committee of the University of Cologne (#13-091). All breast cancer tissues analyzed were from female patients of Caucasian background with an average age of 52.

Tissue staining was carried out as described above. For quantifications based on staining of sections, 5 to 10 different areas were imaged, counted and averaged for each animal. To specifically count the percentage (%) of epithelial tumor cells (Fig. 3a, b and Supplementary Fig. 5a, b), the pH3[+]/E-cad[+], cyclin D1[+]/E-cad[+], clCasp3[+]/E-cad[+] cell numbers were divided by the DAPI/E-cad[+] cell numbers.

Tumoroids attached to the glass bottom were processed as described for cell staining. Tumoroids detached from the glass were PFA fixed and then collected in 1.5 ml-Eppendorf tubes for the subsequent staining and washing. After DAPI counterstaining, the tumoroids were transferred into 8-well Ibidi μ-Slide (#80826) and covered by mounting medium for imaging.

Images of cells and tissues/tumoroids were collected at RT on a Zeiss (Jena) LSM780 confocal laser scanning microscope equipped with a Zeiss Plan-APO ×100 or ×40 oil immersion objective.

### Pearson's correlation coefficient analysis

Correlation analysis was used to examine the colocalization between KANK1 and pan-cadherin signals in LECs at the BM or in multilayered carcinomas. Confocal images were captured over an area of 354 μm x 354 μm as described above. Individual regions measuring 80 μm x 80 μm were then cropped. Regions of interest (ROIs) were drawn on cells that were either in contact with the BM (outer) or not (inner). The Pearson's correlation coefficient between KANK1 and pan-cadherin signals was calculated within each ROI using ImageJ. A paired *t*-test was used to compare matched outer and inner ROIs within the same region. Data were collected from 16 regions across 5 images.

### Flow cytometry (integrin expression profiling)

Cells were trypsinized, washed with PBS and then stained with fluorescent-dye-conjugated integrin antibodies (1:400 in PBS with 1% BSA) for 30 min in dark (on ice). Stained cells were washed twice with PBS before submitted for integrin surface level measurement. The flow cytometry was performed on the BD LSRFortessa X-20 Cell Analyzer and the data was analyzed with Flowjo (version 10.10) software.

## Immunoprecipitation and GFP/mCherry pulldown

For immunoprecipitation (IP) of GFP-tagged or mCherry-tagged proteins, cells were lysed in ice cold lysis buffer (10 mM Tris, 150 mM NaCl, 0.5 mM EDTA and 0.6% NP-40 supplemented with proteinase and phosphatase inhibitors). The lysates were incubated with GFP-Trap® Agarose or RFP-Trap® Agarose from Chromotek for 3 h in cold room. The beads were washed and resuspended following the manufacturers' protocols.

For the IP of endogenous KANK1, the anti-KANK1 antibody (Sigma, HPA056090) was conjugated with Protein A/G Agarose beads (Santa Cruz, SC-2003) in 3% BSA for 1 hour at 4 °C before incubating with the cell lysate as described above. IPs with rabbit IgG (ab171870) was used as negative control. The beads were washed 3 times with binding buffer (10 mM Tris, 150 mM NaCl and 0.5 mM EDTA) and eluted in 2X Laemmli buffer at 95 °C for 10 minutes. Samples were analyzed by Western blot.

## Mass spectrometry

Cells were plated in 6-well plates for overnight serum starvation before 10% FBS were applied for 24 hours (3 biological replicates for each group). Whole cell lysates were collected for quantitative MS analysis on QExactive HF mass spectrometer (Thermo Fisher). Data were processed using the label-free quantification (LFQ) algorithm embedded in the MaxQuant software[62] (version 2.0.1.0).

To identify the interacting partners for KANK1-GFP or GFP-NOS1APc, cell lysates were submitted directly for on-beads digestion (3 biological replicates for each group) after anti-GFP IP. The digested peptides were then extracted, purified in StageTips, analyzed in Exploris 480 mass spectrometer (Thermo Fisher Scientific) or timsTOF Pro (Bruker Daltonics). Raw data were processed as described above and were further analyzed with the Perseus software[63]. The interacting proteins displayed in Fig. 4b, Supplementary Fig. 6a, f, and total proteome in Supplementary Fig. 9d, are provided as Source Data attached to each figure. All MS data have been deposited to the ProteomeXchange Consortium via the PRIDE partner repository with the dataset identifier PXD049122.

## Tumoroids culture and time-lapse imaging

The tumoroid culture was performed as described before[64]. Briefly, the carcinoma tumor tissues were dissected from KANK1-WT^PyMT (16-week-old) or KANK1-KO^PyMT mice (21-week-old) with similar tumor weight (2.6–3.3 g). Around 0.4 g were minced by scalpel, digested at 37 °C in 10 ml digestion medium (DMEM/F12 + 10% P/S + 2 mg/ml collagenase (Sigma, C9407) on a shaker for 2 h, then sequentially sheared in a volume of 10 ml, 5 ml using first serological pipettes and then a flamed glass Paster pipette. The cell solution was finally applied to 100 μm cell strainer. The flow through was centrifuged for 5 minutes at 400 rcf. The cell pellet was treated with 2 ml red blood cell lysis buffer for 5 min at RT (Roche, 11814389001), resuspended in 10 ml basal medium (DMEM/F12 + 10% P/S) and finally allowed to adhere to the fibronectin-coated surface (10 μg/ml) at 37 °C for 5 min to remove the adherent stromal fibroblasts. The remaining small tumor clusters were collected, counted and resuspended in 10 mg/ml cold BME type 2 matrix (Trevigen, 3533-010-02). 40 μl matrix-cell mixture with around 150 tumor cell clusters were seeded into the prewarmed 24-well with glass bottom, allowed to solidify, and then incubated with 800 μl complete organoid culture medium for a continuous 5-day culture.

Tumoroid cultures from MCF7 cells were performed similarly. Cell suspension was prepared with 70 μm cell strainer and diluted to a density of 10 cell clusters/μl before they were embedded in cold BME type 2 matrix (Trevigen, 3533-010-02). TAZ expression was induced with 0.5 μg/ml DOX.

Time-lapse live images were acquired using an inverted microscope (Nikon Eclipse Ti2) equipped with an incubator unit with 5% CO2 and humidity. Images were acquired every hour for 5 days and assembled as Movies by Image J. The organoid growth curve was plotted by normalizing the tumoroid sizes at each time point to their original sizes. The tumoroid size was determined by measuring the area of the corresponding tumoroid with Image J.

## Cell culture, transient transfection, stable viral infection and FACS analysis

MCF7 cells were obtained from ATCC (HTB-22) and were cultured in DMEM (Gibco, 31966-021) supplemented with 10% FBS, Penicillin/ Streptomycin (Gibco, 15140-122) and MEM NEAA (Gibco, 11140-35). Transient transfection of plasmids was carried out with Lipofectamine™ 3000 following the manufacturer's instructions (Invitrogen).

For generation of stable cell line, VSV-G pseudotyped retroviral and lentiviral vectors were produced by transient transfection of HEK293T cells (CRL-3216). Viral particles were concentrated from cell culture supernatant as previously described[4] and used for cell infection.

Bafilomycin A1 (BAF) (40 nM in DMSO) and MG132 (1 μM in DMSO) were applied to cells for 14 h, respectively. Cycloheximide (CHX) was applied at 20 μg/ml in DMSO for the experimental times as indicated.

GFP-positive cells were suspended in FACS buffer (2% FCS + 2.5 mM EDTA in PBS) and sorted by FACSAria Cell Sorter.

## Mammosphere formation assay

Sphere formation was performed as described before[65]. Briefly, confluent (80%) monolayers of cells were released with 0.05% Trypsine (Gibco, 15400-054) and 2.5 mM EDTA. Cells suspension then were allowed to pass through the 26 G needles 3 times before they were applied to 40 μm cell strainer. $10^4$ single cells in 2.5 ml culture medium were plated into Ultra-low attachment 6-well plate (Corning, #3471) and cultured for 5 days. Mammospheres were imaged and counted 5 days later. Only spheres >50 μm were counted under a microscope equipped with 2.5X objective (the bottom of 6-well plates were divided into 9 areas to facilitate the counting). Spheres were thereafter collected and dissociated into single cells for a second round of mammosphere formation. Mammosphere formation medium was freshly prepared and contains DMEM/F12 (Invitrogen, 31330-038), 1XP/S, 20 ng/ml EGF (Pepro Tech, AF-100-15), 10 ng/μl bFGF (Pepro Tech, AF-100-18B) and 1X B27 (Gibco, 17504044). TAZ expression was induced with 0.5 μg/ml DOX.

## MCF7 cell fat pad xenograft

NOD-SCID female mice age of 8 weeks were treated with a low dosage of β-estradiol (4 μg/100 μl PBS per animal) via intraperitoneal injection 3 days prior to transplantations of luciferase-labelled MCF7^KANK1-WT and MCF7^KANK1-KO cells (pLenti-EF1a-Luciferase-IRES-Blast-WPRE was a gift from Javier Alcudia, Addgene plasmid # 108542). For each animal, $5 \times 10^6$ cells were resuspended in 100 μl 50% Matrigel (Corning, 354234) and orthotopically injected into one side of the 4th pair of the mouse mammary glands using 25 G needles. To monitor tumor growth via in vivo bioluminescent imaging (IVIS-100 Xenogen, PerkinElmer), 150 mg/kg of D-luciferin (30 mg/ml) in PBS was intraperitoneally injected into tumor-bearing animals. At the experimental end point, all mice were euthanized and tumors were dissected out for further analysis. Animal care and the described tumor experiments were performed in accordance with the protocols approved by the Committee on the Use of Live Animals in Teaching and Research of the University of Hong Kong (CULATR NO: 5842-21).

## Antibodies

See Supplementary Data 1.

## Plasmids

See Supplementary Data 2.

## Quantitative real time-PCR (qRT-PCR)

Total RNA was extracted from cultured MCF7 cells with RNeasy Mini extraction kit (Qiagen, 74104) and transcribed into cDNA with the iScript cDNA Synthesis Kit (Biorad, 170-8891). Real time PCR was performed with the LightCycler480 Instrument II (Roche). PCR protocol: 5 min 95 °C; 40 x (15 s 95 °C, 15 s 64 °C); 15 s 95 °C; 15 s 60 °C; 15 s 95 °C; ∞ 4 °C. Samples were measured in triplicates. PCR primers are listed in Supplementary Data 3.

## Isoform specific NOS1AP antibody production

Peptide sequences synthesized in the in-house peptide synthesis core facility to generate the isoform specific antibodies are listed in Supplementary Data 4. Peptide synthesis, rabbit immunization and final serum purification were carried out as described previously[10].

## Crispr-Cas9 knockout cell line generation

To disrupt the *KANK1* and *NOS1AP* genes in MCF7 cells, respectively, guide RNA sequences (gRNAs) targeting the corresponding human genes were synthesized and cloned into pSpCas9(BB)−2A-Puro (PX459) V2.0[66]. The *KANK1*-gRNA (GACATCGTCGTGTACCACAG) and *NOS1AP*-gRNA (GAAGGCGTCCTCGTTGTGCA) plasmids were electroporated at a final concentration of 4 μg into $10^6$ cells in a volume of 100 ul using Lonza Nucleofector IIb device with Cell Line Nucleofector Kit V (Lonza#VCA-1003) and program P-020. Puromycin at 3 μg/ml was used to select for positive clones. The frame shift of the corresponding genes was verified by sequencing. At least 2 independent clones were analyzed for each knockout line to exclude clonal effects. Unless stated otherwise, the data shown show the analysis and quantification from Clone#1 of each knockout line. Rescue assays were performed in KANK1-KO Clone#1. Primers for amplification of the Cas9 targeting regions are listed in Supplementary Data 5.

## siRNA-mediated mRNA depletion

siRNA for human NOS1AP and non-targeting pool (control siRNA) were obtained from Dharmacon™ (NOS1AP: L-013446-00-0005 and control: D-001810-10-05). siRNA for Human SCRIB was synthesized by Eurofins (sequence: GUCAUUGGAACAGGACGCU-TT)[37]. 6 nM of each siRNA was delivered into the MCF7 cells in 6-well plates with Lipofectamine™ RNAiMAX (Invitrogen: 13778075-150) according to the manufacturer's protocol.

## Correlation analysis on human data

The TCGA patient cohort for Kaplan-Meier survival analysis was based on the GDC TCGA Breast Cancer (BRCA) (dataset ID: TCGA-BRCA.htseq_fpkm-uq.tsv, version 07-18-2019)[23]. Complete subtype information was extracted from previous publications[67,68] and combined as Supplementary Data 6. Proteomic data extracted from the publication[69] were used for correlation analysis between *KANK1* RNA and protein levels. Data for KANK1 and TAZ subcellular protein abundance was extracted from publication[56]. Source Data used for plotting are provided as Source Data files.

## Statistics & reproducibility

Statistical analysis was carried out in GraphPad Prism Software (version 9). The specific tests and the sample sizes are indicated in the figure legends accompanied with corresponding Source Data.

## Reporting summary

Further information on research design is available in the Nature Portfolio Reporting Summary linked to this article.

## Data availability

Source Data are provided with this paper. The original videos for still-images, the FACS data and the uncropped versions of all blots are also provided. The mass spectrometry proteomics data have been deposited to the ProteomeXchange Consortium via the PRIDE partner repository with the dataset identifier PXD049122.

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

## Acknowledgements

We thank Elizabeth Engle (Harvard Medical School, USA) for KIF21A antibody and Norikazu Yabuta (Osaka University, Japan) for phospho-LATS antibodies, Barbara Steigenberger for MS support, Martin Spitaler and Giovanni Cardone for help with imaging experiments, Stephan Uebel for peptide synthesis, the Max Planck Biochemistry animal house for mouse husbandry, Peter Krenn for help with cell sorting, Nanpeng Chen for discussions, Daniela Ferretti for bioinformatic analyses, Edmire Shala for domain mapping and Alexander Felber for technical support. The work was supported by Hong Kong Research Grants Council (GRF_17118120) to Bo Gao, CIHR (PJT-183923 and PJT-158738) to James P. Fawcett, and the ERC (grant agreement no. 810104 - Point) and the Max Planck Society to Reinhard Fässler.

## Author contributions

S.S.G. and R.F. conceived the study, designed the experiments and analyzed the data. G.M.W. designed the Crispr-Cas9 targeting constructs, performed the Pearson Correlation analysis and contributed to the statistical analyses. Z.L. and B.G. performed the cancer cell xenografting assays. Z.S. generated the KANK1-KO strain. K.Y. performed FACS analysis and generated TAZ constructs for retroviral cell transductions. J.P.F. generated the NOS1APc-specific antibody. R.B. collected, provided and helped analyzing the human breast cancer samples. S.S.G. and R.F. wrote the manuscript with input from G.M.W., B.G. and J.P.F. All authors read and approved the final manuscript.

## Funding

## Competing interests

The authors declare no competing interests.
