## [Transparent Peer Review file · Nature Communications]

KANK1 promotes breast cancer development by compromising Scribble-mediated Hippo activation

Corresponding Author: Dr Shiny Shengzhen Guo

Version 0:

Reviewer comments:

Reviewer #1

(Remarks to the Author)

The MS by Guo et al., describes the cancer-promoting activities of KANK1, a junctional protein identified by the authors as a promoter of breast cancer growth. Mechanistically, the authors propose a model in which these effects are due to the ability of KANK1 to interfere with the binding of the polarity protein Scribble to another scaffold protein known as NOS1AP, ultimately leading to the inhibition of the Hippo pathway and consequent stabilization and activation of the pro-oncogenic transcriptional regulator TAZ.

This story is potentially interesting, but suffers of several limitations (detailed below) that require to be addressed in full to support this MS for publication:

All the issues with this MS emerge by the lack of functional validation of the pathway (Fig. 7i) the authors propose to explain the phenotype of KANK1 KO in breast cancer cells. What the MS offers are only correlative observations, which may be suggestive of a link between KANK1, NOS1AP, Scribble and TAZ, but are far insufficient to support the claim of causal connections between these proteins. In particular:

- 1) The authors have to show that the effects of KANK1 loss in vivo and in vitro can be reverted by adding back TAZ (perhaps using a Hippo-insensitive form of that protein).
- 2) TAZ is known to be mainly involved in supporting clonogenic and tumor-initiating properties of breast cancer cells, although it seems less involved in regulating cell proliferation per se. KANK1 loss in breast cancer cells is shown to affect tumor growth, but it is unclear if this is related to cell cycle arrest, cell death or to a decrease of cells endowed of tumor initiating capacity. The latter is what should be expected from a factor whose main activity in breast cancer is to sustain TAZ stability. Therefore, the authors have to provide evidence in vitro and in vivo that KANK1 is relevant for the clonogenic and tumor-initiating properties of breast cancer cells. For that, the authors should monitor the emergence of secondary mammospheres from wild-type and KANK1 knockout cells; if a decrease is observed in KANK1 KO cells, then the authors should show that this can be rescued by adding-back TAZ. Along the same line of reasoning, the authors should perform limiting dilution assays to test in vivo the tumor initiating capacities of wild-type and KANK1 knockout cells, and then show that the KANK1 KO phenotype can be reversed by adding-back TAZ.
- 3) A corollary of point 2 above is the fact that KANK1 KO should strongly affect the metastatic proclivity of breast cancer cells. This could be easily monitored with the Polyoma Middle T (PyMT) mouse model the authors used to test the relevance of KANK1 in breast cancer, as that model is known to form spontaneous metastases to the lung at high frequency.
- 4) TAZ is a transcriptional regulator. If KANK1 KO impacts on TAZ, then one should expect to observe a corresponding change in the mRNA (not protein) expression of TAZ target genes (CTGF, CYR61, ANKRD1, for example), which should also be rescued by adding back TAZ.
- 5) The functional relationships between KANK1, NOS1AP and Scribble in the context of breast cancer require extensive validation. In particular, the effects of loss of KANK1 at the biochemical, cell culture and in vivo tumor growth levels should be rescued by concomitant loss of NOS1AP or Scribble in the same cells.

Reviewer #2

(Remarks to the Author)

In the present manuscript, Guo and colleagues reveal the mechanism how KANK1 regulates breast cancer development. Using the Polyoma Middle T (PyMT) driven murine breast cancer model with or without KANK1, they show that KANK1 promotes mammary gland tumor growth. They identify high KANK1 expression exclusively in luminal epithelial cells (LEC) where KANK1 changes its location, from basal side in basement membrane-attached tumor cells to cell-cell junctions of basal membrane-detached tumor cells. In a series of experiments involving the search for KANK1-interacting proteins and analysis of the signaling pathway in mouse tumor cell growth in vivo, in tumoroids and in xenografted human breast cancer cells, the authors demonstrate the tumor promoting function of KANK1. They explain the detailed mechanism of how KANK1 translocation affects tumorigenesis. Namely, KANK1 competes with the polarity and tumor suppressor protein SCRIB for association with NOS1AP and thereby curbs the ability of SCRIB to activate Hippo pathway. The final outcome is stabilization of the nuclear TAZ and enhanced cell proliferation. In conclusion this work explains the mechanism of the role of KANK1 in tumor development in vivo.

This is comprehensive, well documented and well written manuscript that provides mechanistic insight into the role of KANK1 in breast cancer development. It clearly shows the mechanism of how KANK1 acts as a tumor promoting factor in breast cancer and explains why TCGA database indicates that high levels of KANK1 are associate with pure prognosis. Authors use different model systems. They use a PyMT model that faithfully recapitulates breast cancer progression in humans. They also use tumoroids of LECs isolated from mice, a human MCF7 breast cancer cell line that resembles primary PyMT-induced carcinoma cells in situ, while some of the conclusions were confirmed in epithelial cells from human kidney embryo HEK293T. MCF7 cells were used in co-immunoprecipitation and Mass Spectrometry analysis, but also for the formation of xenografts in mice. The methods are extremely diverse, clearly described and well adapted to the research objective. In the Materials and Methods section, very detailed information is given about the antibodies used (Supplementary Table 1.), and the representations of the plasmid constructs used (Supplementary Table 2.) are very clear. The images are of excellent quality, and their descriptions provide enough information for the interpretation of the results. The scheme of the entire mechanism is very clear and informative and contributes to easier reading of the paper. The supplementary figures are very comprehensive and full of data that contribute to the understanding and reliability of the results presented in the main text. The statistics used in the article are appropriate for the data presented. Error bars and probability values are provided for all experiments. The interpretation of the data is clear, and the conclusions drawn from it are robust, valid and reliable. They used a large number of controls and the results are very convincing. The bibliography contains articles that are relevant to the researched topic. For these reasons, I strongly support the acceptance of this paper.

There is only one minor point that may be addressed in the discussion:

Could the authors briefly comment on possible changes in the integrin repertoire and/or BM composition during breast cancer development and at least hypothesize what would be the main trigger for cell detachment from the BM.

Reviewer #3

(Remarks to the Author)

The manuscript by Guo et al. clearly elucidates the role of KANK1 in breast cancer promotion, highlighting its novel interaction with NOS1AP and its influence on the Hippo signaling pathway in murine models. The study reveals that KANK1's translocation from focal adhesions to cell-cell junctions disrupts the interaction between SCRIB and NOS1AP. This perturbation subsequently leads to decreased TAZ phosphorylation, nuclear accumulation of TAZ, and ultimately, increased cell proliferation. The authors also demonstrate that this molecular pathway is conserved between human and murine systems.

This is an elegant study that uncovers an important signal pathway, and I am enthusiastic about the work. The manuscript was written in simple, unambiguous and clear language despite some typo. All the experiments are well done. The methods are described in sufficient detail. The results are presented in a coherent and logical sequence, with data interpretation that is both transparent and convincing.

The paper's most significant contribution lies in its pioneering discovery of KANK1's involvement in the Hippo pathway, a previously established factor in the context of focal adhesions, now implicated in breast cancer progression. This revelation is poised to spark widespread interest and further investigation within the scientific community.

Some minor points:

- 1, line 95, the manuscript fails to include the antibody information for KANK3 and 4, as referenced in Figs. S1d and f, within Table 1. This omission should be rectified.
- 2, The manuscript should address an inconsistency on line 95, where the labels atop the middle panel of Fig. S1d do not align with the color-coding. Since Fig. S1 precedes Fig. 2, it would benefit the readers if the authors clarified the significance of the markers Nidogen, pan-can, and CK8 in the legend for Fig. S1. Additionally, the pattern of SMA in fig S1C compared with in S1e&f are different, why?
- 3, line 123, "The KANK1 protein switched from the predominantly basal side of BM-attached tumor cells (arrowhead in Fig. 2d) to the pan-cadherin+ cell cortex of BM-detached tumor cells." However, in Fig. 2d, this conclusion regarding the localization shift of KANK1 is not evident and requires further clarification or evidence.
4. line 169, the protein levels of KANK1 in Fig. 3C appear significantly lower than those of KANK2 or 3. How did the authors confirm the identity of the KANK1 bands? Referring to the endogenous KANK1 (150kd) indicated by the authors in Fig 3d, it

seems that the GFP-KANK1 in Fig 3c have a higher molecular weight (>200kd?), which warrants confirmation.

5. line 184, from the co-IP data in Fig. S3f, it was hardly observed the interaction between CC-UC and UC to NOS1AP, however in Fig 4a marked as (+). This finding requires further elaboration or consideration.

6. line 245, "pS89-TAZ leads to 14-3-3 binding and cytoplasmic retention, and pS311-TAZ to proteasomal degradation." Given that the phosphorylation of S89 and S311 can result in either the stabilization or degradation of TAZ, how to understand that both levels of pS89 and pS311 are reduced in MCF kank1-wt cells?

7. line 279, there appears to be a typographical error: "They had KANK1 at their basal side and TAZ predominantly in the cytoplasm." It is likely that TAZ should be noted as being predominantly in the nucleus, not the cytoplasm.

8. line 798, there is a labeling error within the manuscript: "(c,d) Hematoxylin-eosin H/E staining (b) and grading (c) of," where the labels (b) and (c) should correspond to (c) and (d), respectively. Similar incorrect labeling is also present in lines 809 and 810 and should be corrected for clarity and accuracy.

Reviewer #4

(Remarks to the Author)

The article "KANK1 inhibits the tumor suppressor Scribble by 1 competing for NOS1AP binding" by Guo and colleagues experimentally validates a role for KANK1 in tumor growth using cell culture and in vivo breast cancer models. Deletion of KANK1 results in reduced tumor growth, associated with lower proliferation rates. The authors demonstrate that KANK1 functions by localizing to the lateral plasma membrane in malignant cells and competes with SCRIB for NOS1AP, resulting in activation of TAZ. The competition between KANK1 and SCRIB for NOS1AP a very interesting mechanism and well supported by the data. However, the connection between KANK1 localization and Taz function in promoting proliferation to drive tumor growth is correlative. It would be necessary for the authors to provide a functional link to fully support the conclusions that recruitment of KANK1 to lateral junctions promotes tumour growth through Taz activity. Moreover, the expression of KANK1 is absent from normal mammary gland then present in the malignant cells, but the timing of this transition is not addressed adequately.

Furthermore, the conclusion that KANK1 translocated from focal adhesions to lateral junctions is not directly demonstrated. This conclusion could be supported by additional experiments, or the description could be refined to reflect a different localization pattern in malignant cells that is not necessarily dependent on translocation. Finally, there are several instances where additional methodological information is needed to fully understand the results.

Specific points for consideration are listed below.

Major points:

1. The end of first paragraph of introduction is long and complicated. This should be revised. Additionally, NOS1APc function and role should be included in the introduction to highlight its importance in the hippo pathway. More information that supports KANK1 as a tumor suppressor should also be included in the introduction. It is only briefly mentioned at the beginning of the Results.
2. Fig1. The purpose of the non-breast cancers isn't clear from the stated objective. More information about the datasets and how the TCGA cohorts were chosen should be included in the methods. Data split by breast cancer subtype would also be informative and important, particularly since the experimental models represent luminal (ER+) breast cancer. Figure 1 would further be enhanced by including protein expression from breast tissues. Is high KANK1 restricted to the epithelial compartment in cancers? Do protein and mRNA correlate?
3. For the KANK1-KO, exons 6 and 7 were removed. Is a truncated protein expected? This could be validated by immunoblot with an antibody that recognizes the N-term.
4. Lines 123, 310 and others - The authors claim that KANK1 shifts localization from predominantly basal to cell-cell junction. From the data presented it appears that the basal level is maintained, but it is expressed at cell-cell contacts in multilayered regions that lack contact with the basement membrane. Is this accurate? Figure 2d- Quantification of KANK1 localization considering contact with the basement membrane would be helpful to illustrate if KANK1 is mislocalized in cells not contacting the basement membrane.
5. It appears that KANK1 is not expressed in the virgin gland, but is expressed in transformed cells. It is important to demonstrate when during tumor progression KANK1 starts to be expressed. Does it only occur in stratified ducts? Does it stay basal in cells in contact with basement membrane? (e.g. what does it look like in in situ tissue?). The authors describe translocation when cells detach from the BM - I don't see detachment from the basement membrane per se, but instead cells appear growing with no basement membrane contact in stratified regions. Does this occur during development in end buds, which have similar morphology and increased proliferation to transformed tissue? None of the experiments directly show a translocation from BM to cell-cell contact. Could KANK1 expression in cells with no basement membrane contact go to an alternative site rather than translocation? This could be resolved by looking at earlier tumor samples to determine when KANK1 is expressed.
6. What age are the virgin glands from? KANK1 in puberty development should also be included. Related to this previous point, during pubertal development, the terminal endbud is multilayered and proliferative. Is KANK1 expressed at cadherin+ junctions at this stage?
7. Figure 2d-g. The authors should clarify when the tumors were collected for staining and analysis and if the time points are the same for KANK1-KO and WT. It should also be added in the method section at which time point/tumor volume tumors were collected for organoid culture.
8. In Sup Fig 2c, is KANK1 KO really at palpation? It looks like normal ducts. What was palpated?
9. Fig Supp 2d, how was % calculated?
10. Fig4a- The binding data or quantification of binding mutants should be in Fig 4, not just the schematics and qualitative symbols. The overexposed blot should also be shown. Pull down efficiency from multiple blots of each should be quantified in a graph.
11. Does KANK1-del-UC-GFP fail to localize to junctions, as would be expected in experiments similar to Fig 4d?

12. Fig4d- object correlation of KANK1 and paxillin should be quantified.
13. Line 250 - the authors claim that total Taz levels increase with serum, but this is not apparent from Fig 6 e and f. Furthermore, for Fig6-e - the authors should show quantification of these blots. Clearer interpretation and explanation of this panel should be provided as results aren't clear from the blot itself.
14. Line 254 - the authors claim higher activity of LATS, explanation and quantification should be provided as an increase in phosphorylation activity isn't clear from the blot.
15. The rescue of KANK1-KO with KANK1-GFP-del-UC would help strengthen evidence that binding to NOS1AP by KANK1 is necessary for the effect on Taz in supp Fig 6c.
16. The authors show a strong correlation between KANK1 expression and Taz localization in tumors, but it is necessary for the authors to demonstrate that the growth effect of KANK1-KO is indeed dependent on Taz signaling. This could be achieved by expressing a Taz mutant that is not phosphorylated and degraded.
17. Fig 7a shows high levels of cytoplasmic Taz in the KANK1-KO samples. However, this appears to contradict experiments in Fig 6 showing that Taz is degraded in KANK1-deficient conditions.
18. The results would be further strengthened by providing validation in human patient breast cancer samples. Is there a correlation between KANK1 and nuclear Taz? Is there a correlation between KANK1 expression and Taz target genes (e.g. CTGF) in human breast cancers (e.g. TCGA)? This would be expected from the conclusions presented.
19. The key aspect of the paper seems to be the inappropriate expression of KANK1 in LEC during cancer progression. How KANK1 is expressed (or when) is important to the mechanism but is not addressed experimentally or in the discussion.

Minor points:

1. The authors should avoid red/green images to provide figures that are more accessible to colorblind individuals.
2. Line 94 - it is not accurate that breast cancer arises from pregnancy and lactation luminal epithelial cells.
3. Fig 2c – it would be easier to compare with 2a and b if it was also in weeks. 2b shows that KO at 21 weeks is the same size as WT at 16 weeks. This is not apparent in 2c.
4. Line 183 - "interacting domain" - UC-Ank is not really a domain, but a region of unstructured adjacent to the Ank repeats.
5. Should Supp Fig 4b and c have Upper and Bottom labels?
6. Line 91. The justification for looking at KANK1 in breast cancer is not clear and does not correspond to data shown in fig.1. The description of the rationale for looking at KANK1 in breast cancer and could be improved.
7. Fig 2f. The Y axis, # of positive cells/animal is unclear, is it in the mammary gland? Mammary epithelium?

Reviewer #5

(Remarks to the Author)

Version 1:

Reviewer comments:

Reviewer #1

(Remarks to the Author)

In my previous comments, I asked the authors to carry out new in vitro and in vivo experiments to fully support their claim of causal connections between KANK1, NOS1AP, Scribble and TAZ (Fig. 8j).

In answer, the authors provided several new data from experiments carried out in vitro, showing the expected epistatic relationships between KANK1 and the other factors (although the effects appear rather weak, see the graphs on the right of Supp.Fig.11e).

Yet, the authors did not perform any of the in vivo experiments I requested in my points 1, 2, 3 and 5, saying it was because they lacked the necessary permissions. This is apparently true also for my point 3, in which I requested to monitor formation of metastatic nodules on the mice they already used for their initial submission. In that case, the authors have the permission to work with tumor-developing transgenic mice, but that permission does not allow mice reaching a stage in which rampant metastases would form.

I sympathize with the authors' struggle with the roadblocks posed by the EU against animal experimentation; nonetheless, I still think that the main claim of this MS needs some in vivo validation, perhaps through collaborations with some other laboratories outside EU. In particular, it would be important to validate that TAZ works as the downstream effector of KANK1 to foster tumor growth, showing that Hippo-resistant TAZ can rescue the effects of KANK1 loss. Without that, the claim of a new oncogenic pathway involved in the biology of breast cancer would be rather weak, as it would rely only on in vitro experiments showing weak effects. As results, the story would result poorly attractive for general readers and readers interested in cancer biology.

Reviewer #2

(Remarks to the Author)

After carefully considering the authors' responses to both my comments and those of the other reviewers, I find the revisions satisfactory. The authors have adequately addressed the concerns raised in the initial review, and the manuscript has been significantly improved. I recommend its acceptance in Nature Communications without further revision.

Andreja Ambriović-Ristov

Reviewer #3

(Remarks to the Author)

I am satisfied with the revision.

Reviewer #4

(Remarks to the Author)

The authors have effectively addressed the points raised in the initial review and the revisions made to the manuscript are both thorough and satisfactory. The revisions have significantly enhanced the clarity of the manuscript.

However, it remains unclear from the data presented in Figure 7e and h that LATS activity is higher in KANK1-KO. While it's agreed that the ratio is increased based on normalized data (i.e. the proportion of LATS that is phosphorylated has increased), the total LATS activity appears unchanged. Nonetheless, this is not a major point and should not impede acceptance of an otherwise very solid manuscript.

Minor point:

Fig1 B – Typo in “Subtype”

Reviewer #5

(Remarks to the Author)

POINT-BY-POINT REBUTTAL TO REVIEWERS' COMMENTS

Reviewer #1 (Remarks to the Author):

The MS by Guo et al., describes the cancer-promoting activities of KANK1, a junctional protein identified by the authors as a promoter of breast cancer growth. Mechanistically, the authors propose a model in which these effects are due to the ability of KANK1 to interfere with the binding of the polarity protein Scribble to another scaffold protein known as NOS1AP, ultimately leading to the inhibition of the Hippo pathway and consequent stabilization and activation of the pro-oncogenic transcriptional regulator TAZ. This story is potentially interesting, but suffers of several limitations (detailed below) that require to be addressed in full to support this MS for publication:

All the issues with this MS emerge by the lack of functional validation of the pathway (Fig. 7i) the authors propose to explain the phenotype of KANK1 KO in breast cancer cells. What the MS offers are only correlative observations, which may be suggestive of a link between KANK1, NOS1AP, Scribble and TAZ, but are far insufficient to support the claim of causal connections between these proteins. In particular:

1) The authors have to show that the effects of KANK1 loss in vivo and in vitro can be reverted by adding back TAZ (perhaps using a Hippo-insensitive form of that protein).

-This is an important request. We retrovirally transduced MCF7^{KANK1-KO} cells with doxycycline-inducible HA-tagged Hippo insensitive TAZ-S89A/S311A mutant (TAZ^{SSAA}) and monitored tumoroid growth over a 5-day period by time-lapse live-imaging. As predicted by the reviewer, TAZ^{SSAA} reconstitution reversed the growth defects observed in MCF7^{KANK1-KO} cells. The result is shown as Supplementary Fig. 11b-c with revised text at line 339-341.

-Unfortunately, we were unable to carry out the in vivo experiments in a reasonable time frame. The experiments require approval by the local government. We would need to submit a new application, which will take at least one year in Bavaria/Germany. Therefore, we decided to test the Hippo-insensitive TAZ in tumoroid experiment.

2) TAZ is known to be mainly involved in supporting clonogenic and tumor-initiating properties of breast cancer cells, although it seems less involved in regulating cell proliferation per se. KANK1 loss in breast cancer cells is shown to affect tumor growth, but it is unclear if this is related to cell cycle arrest, cell death or to a decrease of cells endowed of tumor initiating capacity. The latter is what should be expected from a factor whose main activity in breast cancer is to sustain TAZ stability. Therefore, the authors have to provide evidence in vitro and in vivo that KANK1 is relevant for the clonogenic and tumor-initiating properties of breast cancer cells. For that, the authors should monitor the emergence of secondary mammospheres from wild-type and KANK1 knockout cells; if a decrease is observed in KANK1 KO cells, then the authors should show that this can be rescued by adding-back TAZ. Along the same line of reasoning, the authors should perform limiting dilution assays to test in vivo the tumor initiating capacities of wild-type and KANK1 knockout cells, and then show that the KANK1 KO phenotype can be reversed by adding-back TAZ.

We addressed the tumor-initiating capacity by comparing the emergence of primary and secondary mammospheres generated with MCF7^{KANK1-WT}, MCF7^{KANK1-KO} and MCF7^{KANK1-KO+TAZ(SSAA)} cells.

-In line with published findings showing that TAZ endows self-renewal potential of breast cancer cells, we observed that MCF7^{KANK1-KO} cells with their reduced TAZ activity displayed a decreased tumor-initiating capacity characterized by fewer and smaller mammospheres. Importantly, expression of the Hippo-insensitive TAZ^{SSAA} mutant re-established the tumor initiating capacity in MCF7^{KANK1-KO} cells. The results are shown in the revised Supplementary Fig. 11d-f with revised text at line 342-348.

-As mentioned above, in vivo limiting dilution assays requires approval by the local government and hence the submission of a new application, which is a long-lasting process in Germany (and probably in the entire European Union and not only in Germany).

3) A corollary of point 2 above is the fact that KANK1 KO should strongly affect the metastatic proclivity of breast cancer cells. This could be easily monitored with the Polyoma Middle T (PyMT) mouse model the authors used to test the relevance of KANK1 in breast cancer, as that model is known to form spontaneous metastases to the lung at high frequency.

We studied primary tumor growth in mice with C57BL/6 background. Although the in vivo experiments were approved by the local government authorities, we were only allowed to keep tumor mice alive that displayed palpable tumor size of up to '3 cm diameter in sum'. Only few of the many mice we analyzed, develop lung metastasis within the approved time frame. We plan to analyze metastasis in future using the FVB background, which suffers from higher lung metastasis rate (PMID: 33235291). We are currently applying for these experiments.

4) TAZ is a transcriptional regulator. If KANK1 KO impacts on TAZ, then one should expect to observe a corresponding change in the mRNA (not protein) expression of TAZ target genes (CTGF, CYR61, ANKRD1, for example), which should also be rescued by adding back TAZ.

-The mRNA expression levels of TAZ targets in serum-starved cells were analyzed at Day-1 and Day-3 after serum treatment. The results of these experiments are shown in Supplementary Fig. 8a with the revised text at line 306-309. Both CTGF and CYR61 expression levels were decreased upon loss of KANK1 expression.

-ANKRD1 expression level is very low in MCF7 cell line and was therefore, excluded from the analysis.

-Expression of TAZ^{WT} or TAZ^{SSAA} in MCF7^{KANK1-KO} cells restored CTGF and CYR61 expression levels at Day-1. The rescue effects were sustained at Day-3 in TAZ^{SSAA} but not in TAZ^{WT} expressing cells, suggesting that TAZ stabilization is the key event regulated by KANK1 (please consult revised Supplementary Fig. 11a and revised text at line 327-339).

5) The functional relationships between KANK1, NOS1AP and Scribble in the context of breast cancer require extensive validation. In particular, the effects of loss of KANK1 at

the biochemical, cell culture and in vivo tumor growth levels should be rescued by concomitant loss of NOS1AP or Scribble in the same cells.

We investigated the functional relationship between the three players by siRNA depletion of either NOS1AP (Supplementary Fig. 10a-b) or Scribble (Supplementary Fig. 10c-d) in MCF7^{KANK1-KO} cells. The experiments revealed that TAZ stability was regained upon depletion of either NOS1AP or Scribble, which was further supported by functional tests, in which we expressed the Hippo-insensitive TAZ^{SSAA} during tumoroid growth (Supplementary Fig. 11b-c) and mammosphere formation (Supplementary Fig. 11d-f). The revised text can be found at line 323-326.

Reviewer #2 (Remarks to the Author):

In the present manuscript, Guo and colleagues reveal the mechanism how KANK1 regulates breast cancer development. Using the Polyoma Middle T (PyMT) driven murine breast cancer model with or without KANK1, they show that KANK1 promotes mammary gland tumor growth. They identify high KANK1 expression exclusively in luminal epithelial cells (LEC) where KANK1 changes its location, from basal side in basement membrane-attached tumor cells to cell-cell junctions of basal membrane-detached tumor cells. In a series of experiments involving the search for KANK1-interacting proteins and analysis of the signaling pathway in mouse tumor cell growth *in vivo*, in tumoroids and in xenografted human breast cancer cells, the authors demonstrate the tumor promoting function of KANK1. They explain the detailed mechanism of how KANK1 translocation affects tumorigenesis. Namely, KANK1 competes with the polarity and tumor suppressor protein SCRIB for association with NOS1AP and thereby curbs the ability of SCRIB to activate Hippo pathway. The final outcome is stabilization of the nuclear TAZ and enhanced cell proliferation. In conclusion this work explains the mechanism of the role of KANK1 in tumor development *in vivo*.

This is comprehensive, well documented and well written manuscript that provides mechanistic insight into the role of KANK1 in breast cancer development. It clearly shows the mechanism of how KANK1 acts as a tumor promoting factor in breast cancer and explains why TCGA database indicates that high levels of KANK1 are associated with pure prognosis.

Authors use different model systems. They use a PyMT model that faithfully recapitulates breast cancer progression in humans. They also use tumoroids of LECs isolated from mice, a human MCF7 breast cancer cell line that resembles primary PyMT-induced carcinoma cells *in situ*, while some of the conclusions were confirmed in epithelial cells from human kidney embryo HEK293T. MCF7 cells were used in co-immunoprecipitation and Mass Spectrometry analysis, but also for the formation of xenografts in mice. The methods are extremely diverse, clearly described and well adapted to the research objective. In the Materials and Methods section, very detailed information is given about the antibodies used (Supplementary Table 1.), and the representations of the plasmid constructs used (Supplementary Table 2.) are very clear. The images are of excellent quality, and their descriptions provide enough information for the interpretation of the results. The scheme of the entire mechanism is very clear and informative and contributes to easier reading of the paper. The supplementary figures are very comprehensive and full of data that contribute to the understanding and reliability of the results presented in the main text. The statistics used in the article are appropriate for the data presented. Error bars and probability values are provided for all experiments. The interpretation of the data is clear, and the conclusions drawn from it are robust, valid and reliable. They used a large number of controls and the results are very convincing. The bibliography contains articles that are relevant to the researched topic. For these reasons, I strongly support the acceptance of this paper.

There is only one minor point that may be addressed in the discussion:

Could the authors briefly comment on possible changes in the integrin repertoire and/or BM composition during breast cancer development and at least hypothesize what would be the main trigger for cell detachment from the BM.

-The cell surface expression profile of integrins does not change upon KANK1 loss (see Supplementary Fig. 7b with revised text at line 282-283).

-During tumor growth, cells lost their cell polarity and grow into the ductal lumen, where they form multiple layers in an ECM-attachment independent (anoikis-resistant) manner (PMID: 35829806). We do not know how tumor cells manage to detach from the BM. It is possible that integrin inactivation during M-phase (PMID: 35469017) combined with cell crowding at the BM underlies the detachment and stratification.

Reviewer #3 (Remarks to the Author):

The manuscript by Guo et al. clearly elucidates the role of KANK1 in breast cancer promotion, highlighting its novel interaction with NOS1AP and its influence on the Hippo signaling pathway in murine models. The study reveals that KANK1's translocation from focal adhesions to cell-cell junctions disrupts the interaction between SCRIB and NOS1AP. This perturbation subsequently leads to decreased TAZ phosphorylation, nuclear accumulation of TAZ, and ultimately, increased cell proliferation. The authors also demonstrate that this molecular pathway is conserved between human and murine systems.

This is an elegant study that uncovers an important signal pathway, and I am enthusiastic about the work. The manuscript was written in simple, unambiguous and clear language despite some typo. All the experiments are well done. The methods are described in sufficient detail. The results are presented in a coherent and logical sequence, with data interpretation that is both transparent and convincing.

The paper's most significant contribution lies in its pioneering discovery of KANK1's involvement in the Hippo pathway, a previously established factor in the context of focal adhesions, now implicated in breast cancer progression. This revelation is poised to spark widespread interest and further investigation within the scientific community.

Some minor points:

1, line 95, the manuscript fails to include the antibody information for KANK3 and 4, as referenced in Figs. S1d and f, within Table 1. This omission should be rectified.

Thank you for indicating this mistake! The omission is included in the revised Table 1.

2, The manuscript should address an inconsistency on line 95, where the labels atop the middle panel of Fig. S1d do not align with the color-coding. Since Fig. S1 precedes Fig. 2, it would benefit the readers if the authors clarified the significance of the markers Nidogen, pan-can, and CK8 in the legend for Fig. S1. Additionally, the pattern of SMA in fig S1C compared with in S1e&f are different, why?

-Label and color-coding in Fig. S1d has been corrected and updated in the revised Supplementary Fig. 2c.

-The marker information is now included and highlighted in the revised figure legends of Supplementary Fig. 2 at line 991-999.

-The different SMA patterns in the original Figure S1c (Supplementary Fig. 2b in the revised manuscript) versus original Figure S1e,f (Supplementary Fig. 2d,e in the revised manuscript) are due to the sectioning angle. SMA+ myoepithelial cells form extensions that reach between CK8+ LECs. If the sectioning angle does not include these intersections, the SMA signals appears continuous as shown in Supplementary Fig. 2d,e. We purposely searched for section angle lacking these extensions to better highlight the absence of KANK2 and KANK4 in LECs.

3, line 123, "The KANK1 protein switched from the predominantly basal side of BM-

attached tumor cells (arrowhead in Fig. 2d) to the pan-cadherin+ cell cortex of BM-detached tumor cells." However, in Fig. 2d, this conclusion regarding the localization shift of KANK1 is not evident and requires further clarification or evidence.

-The arrow heads in Fig. 2d show KANK1 enriched at the basal side of BM-associated cells and the asterisks show KANK1 at cell-cell junctions in cells that grew away from the BM. The co-localization of KANK1 and Pan-Cadherin in stratified carcinoma cells that are not in contact with BM is also visible in Fig. 2e.

-We also provide new KANK1 immuno-staining of mammary tumors harvested 10 or 59 days after they were palpated (see revised Supplementary Fig. 3e and text at line 153-158). In these immunostaining KANK1 is present at the basal side in early-transformed LECs that were still associated with the BM and at cell-cell junction of carcinoma cells that grew into the lumen and stratified.

4. line 169, the protein levels of KANK1 in Fig. 3C appear significantly lower than those of KANK2 or 3. How did the authors confirm the identity of the KANK1 bands? Referring to the endogenous KANK1 (150kd) indicated by the authors in Fig 3d, it seems that the GFP-KANK1 in Fig 3c have a higher molecular weight (>200kd?), which warrants confirmation.

-We think this is due to the fact that the larger the plasmid is the lower the transfection rate, and hence the lower the expression level (PMID: 15336645 and 27918590).

-The KANK1 identity was confirmed by re-staining the same membrane with GFP antibody and KANK1 antibody, indicating the specific band above 150 in the lane transfected with KANK1-GFP was indeed KANK1 (see revised Supplementary Fig. 5e, and text at line 207-209).

-In 10% SDS-PAGE gel (home-made), proteins with molecular weights of around 150 and higher cannot be perfectly separated. Hence, the difficulties to exactly determine molecular weights is due to technical limitations. When separating proteins with the commercially available gradient (4-15%) SDS-PAGE (Bio-Rad, Cat:4561086), the molecular weight of endogenous KANK1 is 150 kD (see Fig. 7a,c,e).

For clarity, we provide the gel format in the Data Availability section that contains the original uncropped blots.

5. line 184, from the co-IP data in Fig. S3f, it was hardly observed the interaction between CC-UC and UC to NOS1AP, however in Fig 4a marked as (+). This finding requires further elaboration or consideration.

The interaction of KANK1 CC-UC and UC with NOS1APc is very weak and is only visible when blots are overexposed. We included an overexposed blot in the revised Fig. 4b.

6. line 245, "pS89-TAZ leads to 14-3-3 binding and cytoplasmic retention, and pS311-TAZ to proteasomal degradation." Given that the phosphorylation of S89 and S311 can result in either the stabilization or degradation of TAZ, how to understand that both levels of pS89 and pS311 are reduced in MCF kank1-wt cells?

Both TAZ phosphosites (TAZ-S89 and TAZ-S311) become phosphorylated by LATS, and the levels of both, TAZ-pS89 and TAZ-pS311 are reduced in MCF7^{KANK1-WT} cells. The phosphorylation of both sites results in (1) increased cytoplasmic retention (TAZ-pS89)

and (2) less stabilization (TAZ-pS311). Hence, we suggest that the phosphorylation of both sites contributes to the low nuclear TAZ.

7. line 279, there appears to be a typographical error: "They had KANK1 at their basal side and TAZ predominantly in the cytoplasm." It is likely that TAZ should be noted as being predominantly in the nucleus, not the cytoplasm.

This must be a misunderstanding. When tumor cells are still associated with BM, KANK1 is predominantly at the basal side, away from Scribble, which is at the cell-cell junctions. Consequently, KANK1 cannot disrupt the Scribble-NOS1AP complex, which in turn leads to the retention of TAZ in the cytoplasm.

8. line 798, there is a labeling error within the manuscript: "(c,d) Hematoxylin-eosin H/E staining (b) and grading (c) of," where the labels (b) and (c) should correspond to (c) and (d), respectively. Similar incorrect labeling is also present in lines 809 and 810 and should be corrected for clarity and accuracy.

Thank you for indicating the mistakes. The labellings have been corrected in the revised figure legends (see line 1012 and 1035-1037).

Reviewer #4 (Remarks to the Author):

The article “KANK1 inhibits the tumor suppressor Scribble by competing for NOS1AP binding” by Guo and colleagues experimentally validates a role for KANK1 in tumor growth using cell culture and in vivo breast cancer models. Deletion of KANK1 results in reduced tumor growth, associated with lower proliferation rates. The authors demonstrate that KANK1 functions by localizing to the lateral plasma membrane in malignant cells and competes with SCRIB for NOS1AP, resulting in activation of TAZ. The competition between KANK1 and SCRIB for NOS1AP is a very interesting mechanism and well supported by the data. However, the connection between KANK1 localization and Taz function in promoting proliferation to drive tumor growth is correlative. It would be necessary for the authors to provide a functional link to fully support the conclusions that recruitment of KANK1 to lateral junctions promotes tumour growth through Taz activity. Moreover, the expression of KANK1 is absent from normal mammary gland then present in the malignant cells, but the timing of this transition is not addressed adequately. Furthermore, the conclusion that KANK1 translocated from focal adhesions to lateral junctions is not directly demonstrated. This conclusion could be supported by additional experiments, or the description could be refined to reflect a different localization pattern in malignant cells that is not necessarily dependent on translocation. Finally, there are several instances where additional methodological information is needed to fully understand the results.

Specific points for consideration are listed below.

Major points:

1. The end of first paragraph of introduction is long and complicated. This should be revised. Additionally, NOS1AP function and role should be included in the introduction to highlight its importance in the hippo pathway. More information that supports KANK1 as a tumor suppressor should also be included in the introduction. It is only briefly mentioned at the beginning of the Results.

-The first paragraph of the introduction has been revised (see line 43-49).

-Information regarding the tumor suppressor function of KANK1 was highlighted at line 55-60, and NOS1AP (see line 81-91) in the revised introduction.

2. Fig1. **The purpose of the non-breast cancers isn't clear from the stated objective.** More information about the datasets and how the TCGA cohorts were chosen should be included in the methods. **Data split by breast cancer subtype would also be informative and important, particularly since the experimental models represent luminal (ER+) breast cancer.** Figure 1 would further be enhanced by **including protein expression from breast tissues.** Is high KANK1 restricted to the epithelial compartment in cancers? Do protein and mRNA correlate?

-The in silico non-breast cancer data was taken out from the revised manuscript.

-More information about the datasets on the TCGA as well as for the patient subtypes has been included in the revised Methods Section (see line 612-619).

-Correlation between KANK1 expression and patient survival based on cancer subtypes is shown in revised Fig. 1a-f with revised text at line 115-118.

-Staining of human breast cancer samples are shown in the revised Supplementary Fig. 12a. The staining shows that KANK1 expression is restricted to the epithelial cell compartment and is absent in stroma cells (see revised text at line 361-363).

-KANK1 protein and mRNA expression level have high correlation ($r=0.7055$, $p<0.0001$) in breast tumors. This is shown in the revised Supplementary Fig. 1c, and mentioned in the revised text (see line 115-118).

3. For the KANK1-KO, exons 6 and 7 were removed. Is a truncated protein expected? This could be validated by immunoblot with an antibody that recognizes the N-term.

-Removal of exon 6 and 7 results in premature stop codons in exon 9 leading to non-sense mediated mRNA decay.

-We verified this hypothesis with a commercial polyclonal antibody from Sigma (HPA056090). The antibody detects full-length KANK1 with a MW of ~150 kDa in tissue lysates. In KANK1-KO tissue lysates the antibody does not recognize truncated KANK1 polypeptides. The data are shown in the revised Supplementary Fig. 1b and mentioned in the revised text (see line 102-105).

4. Lines 123, 310 and others - The authors claim that KANK1 shifts localization from predominantly basal to cell-cell junction. From the data presented it appears that the basal level is maintained, but it is expressed at cell-cell contacts in multilayered regions that lack contact with the basement membrane. Is this accurate? Figure 2d- Quantification of KANK1 localization considering contact with the basement membrane would be helpful to illustrate if KANK1 is mislocalized in cells not contacting the basement membrane.

-We agree with the Reviewer that the KANK1 is maintained at the basal side of BM-associated cells and localizes at cell-cell contacts in multilayered cells that grew away from the BM.

-Pearson Correlation analysis demonstrated that KANK1 signals showed a higher correlation with pan-Cadherin signals in stratified cells lacking BM contact than in cells that are still in contact with the BM. The findings are shown in the revised Fig. 2e (see also revised text at line 157-158). The information on how the calculation was performed has been included in the revised Methods Section (see line 477-485).

5. It appears that KANK1 is not expressed in the virgin gland, but is expressed in transformed cells. It is important to demonstrate when during tumor progression KANK1 starts to be expressed. Does it only occur in stratified ducts? Does it stay basal in cells in contact with basement membrane? (e.g. what does it look like in in situ tissue?). The authors describe translocation when cells detach from the BM - I don't see detachment from the basement membrane per se, but instead cells appear growing with no basement membrane contact in stratified regions (a). Does this occur during development in end buds, which have similar morphology and increased proliferation to transformed tissue (b)? None of the experiments directly show a translocation from BM to cell-cell contact. Could KANK1 expression in cells with no basement membrane contact go to an alternative site rather than translocation? This could be resolved by looking at earlier tumor samples to determine when KANK1 is expressed (a).

These are great suggestions.

-To address question (a), we stained tumor tissues 10 and 59 days after the tumors were first palpated. The staining showed that in 10-day tumor samples, KANK1 is not expressed in non-transformed regions, while in transformed region, cells show weak KANK1 signals at the basal side, and at the cell-cell contact in cells growing into the ductal lumen (see revised Supplementary Fig. 3e and text at line 153-158).

-To address question (b) that whether tissue stratification is the trigger for KANK1 expression, we stained end buds in MGs of 5-week-old females (see revised Supplementary Fig. 3f), and found that KANK1 is only expressed at the basal side of cap cells located at the TEB front, and neither in the TEB neck region nor in the multiple-layered body cells. This indicates that stratification is very likely not the trigger for KANK1 expression (see revised text at line 158-167).

6. What age are the virgin glands from? KANK1 in puberty development should also be included. Related to this previous point, during pubertal development, the terminal endbud is multilayered and proliferative. Is KANK1 expressed at cadherin+ junctions at this stage?

-The virgin glands were derived from 4-month-old females to match the end point analyses of the tumor bearing mice (16-week). We mated 4-month-old females for harvesting MGs at pregnancy (E14.5), at lactation (10 days after pups were born) and involution (1 month after weaning). This information has been included in revised Methods Section (see line 460-463).

-MGs at puberty were obtained from 5-week-old female mice. This information has been included into the revised Methods Section (see line 460-463). The staining of KANK1 in multilayered terminal endbuds is shown in the revised Supplementary Fig. 3f (also see line 158-167).

7. Figure 2d-g. The authors should clarify when the tumors were collected for staining and analysis and if the time points are the same for KANK1-KO and WT. It should also be added in the method section at which time point/tumor volume tumors were collected for organoid culture.

-Since KANK1-WT^{PyMT} and KANK1-KO^{PyMT} mice displayed different tumor onset and growth rate, the timepoints for the tumor collections were not always the same. The timepoints and tumor volumes are provided in the Source Data file. The original Fig. 2d-g changed to Fig. 2f,g in the revised manuscript.

-Tumor tissue for organoid culture was harvested from 16-weeks old KANK1-WT^{PyMT} mice and 21-week-old KANK1-KO^{PyMT} mice. They had similar tumor burden. We used 0.4 g tumor tissues for enzymatic digestion and organoid culture. The information is included in the revised Methods Section (see also line 520-521).

8. In Sup Fig 2c, is KANK1 KO really at palpation? It looks like normal ducts. What was palpated?

-The KANK1-WT^{PyMT} tumors can be palpated at day 10, while KANK1-KO^{PyMT} tumors in litter mates have not yet formed. This is the reason why their ducts look normal. We revised the text for clarification (see line 144-146).

9. Fig Supp 2d, how was % calculated?

-The original Supp 2d is now shown as Supplementary Fig. 3d in the revised manuscript. The tumors at different stages were graded by analyzing a total of 12~15 sections. To obtain a good overview of the tumor grade, we collected a total of 120-150 sections from each tumor and graded the sections at an interval of 10. This approach was used to calculate the %. This information is now included in the revised Methods Section (see line 445-451).

10. Fig4a- The binding data or quantification of binding mutants should be in Fig 4, not just the schematics and qualitative symbols. The overexposed blot should also be shown. Pull down efficiency from multiple blots of each should be quantified in a graph.

-We have moved the binding data and the overexposed blot into the main figure. The quantification of binding relative to input is now also shown in the revised Fig. 4b,c and 4e,f.

11. Does KANK1-del-UC-GFP fail to localize to junctions, as would be expected in experiments similar to Fig 4d?

-The KANK1-del-UC-GFP failed to localize to the cell-cell junctions. As expected, however, it still localizes to FAs. The data are shown in the revised Fig. 5b5-b6 (see also revised text at line 252-254).

12. Fig4d- object correlation of KANK1 and paxillin should be quantified.

-Thank you for the suggestion. The line profiling of KANK1 and paxillin signals is shown in the revised Fig. 5b (see also revised text at line 247-254).

13. Line 250 - the authors claim that total Taz levels increase with serum, but this is not apparent from Fig 6 e and f. Furthermore, for Fig6-e - the authors should show quantification of these blots. Clearer interpretation and explanation of this panel should be provided as results aren't clear from the blot itself.

-The comparison of TAZ levels between starved and serum-treatment cells is shown in the revised Fig. 7f, with p values labelled at the lateral sides. The description can be found in the revised text at line 300-302. We observed that TAZ level was stabilized in MCF7^{KANK1-WT} cells but decreased in MCF7^{KANK1-KO} cells.

-The quantification along with the blots is shown in the revised Fig. 7e-i.

14. Line 254 - the authors claim higher activity of LATS, explanation and quantification should be provided as an increase in phosphorylation activity it isn't clear from the blot.

-In the absence of KANK1, the SCRIB/NOS1AP complex is able to activate LATS (PMID: 22078877 and 25918243).

-Although the pLATS levels are comparable between MCF7^{KANK1-WT} and MCF7^{KANK1-KO} cells, normalization to total LATS levels (which are reduced in MCF7^{KANK1-KO} cells) resulted in higher pLATS activity. The quantification is shown in Fig. 7h.

15. The rescue of KANK1-KO with KANK1-GFP-del-UC would help strengthen evidence that binding to NOS1AP by KANK1 is necessary for the effect on Taz in supp Fig 6c.

-KANK1-ΔUC-GFP was lentivirally transduced into MCF7^{KANK1-KO} cells and nuclear translocation of TAZ was then examined. The result is shown in the revised Supplementary Fig. 9c,d. Consistent with the mapping results showing there is no interaction between KANK1-ΔUC-GFP and NOS1AP, KANK1-ΔUC-GFP failed to localize at the cell-cell junctions (see revised Fig. 5b5-b6) and to rescue the impaired TAZ nucleus localization in MCF7^{KANK1-KO} cells (see revised text at line 315-316).

16. The authors show a strong correlation between KANK1 expression and Taz localization in tumors, but it is necessary for the authors to demonstrate that the growth effect of KANK1-KO is indeed dependent on Taz signaling. This could be achieved by expressing a Taz mutant that is not phosphorylated and degraded.

-This is a great suggestion! We expressed non-phosphorylatable TAZ (S89A/S311A, TAZ^{SSAA}) in MCF7^{KANK1-KO} cells and monitored the growth of tumoroids by time-lapse live-imaging (see revised Supplementary Fig 11b-c). The experiment revealed that overexpression of TAZ^{SSAA} rescued the growth defects in MCF7^{KANK1-KO} cells and upregulated expression of TAZ target genes (see revised Supplementary Fig. 11a and text at line 327-341).

17. Fig 7a shows high levels of cytoplasmic Taz in the KANK1-KO samples. However, this appears to contradict experiments in Fig 6 showing that Taz is degraded in KANK1-deficient conditions.

-The immuno-fluorescence signals of TAZ in the originally submitted manuscript (Fig. 7a - now Fig. 8a in the revised manuscript) are suited to localize but not to quantify TAZ. The western blot signals (Fig. 7e-g) are suited to quantify but not to localize TAZ.

18. The results would be further strengthened by providing validation in human patient breast cancer samples. Is there a correlation between KANK1 and nuclear Taz? Is there a correlation between KANK1 expression and Taz target genes (e.g. CTGF) in human breast cancers (e.g. TCGA)? This would be expected from the conclusions presented.

-The correlation between KANK1 and nuclear TAZ in human breast cancer samples is shown by immuno-fluorescence staining in Fig. 8f. Furthermore, we bioinformatically confirmed this finding by correlating the subcellular localization of membrane localized KANK1 and nucleus TAZ using published data sets (Fig. 8g). The text is updated at line 363-367.

-We also analyzed the correlation between KANK1 and TAZ target genes expression (CTGF and CYR61) using TCGA database, which is shown in the revised Supplementary Fig. 12b-g (see also revised text at line 367-369).

19. The key aspect of the paper seems to be the inappropriate expression of KANK1 in LEC during cancer progression. How KANK1 is expressed (or when) is important to the mechanism but is not addressed experimentally or in the discussion.

-It was shown that KANK1 expression can be induced by progestin treatment (PMID: 24897521), which hints to hormone-induced KANK1 expression in pregnancy and lactation. We do not know how KANK1 expression is upregulated upon cell transformation. This is a very interesting question that will be analyzed in future.

Minor points:

1. The authors should avoid red/green images to provide figures that are more accessible to colorblind individuals.

The colors have been adjusted in the revised manuscript.

2. Line 94 - it is not accurate that breast cancer arises from pregnancy and lactation luminal epithelial cells.

This wrong statement is deleted in the revised manuscript.

3. Fig 2c – it would be easier to compare with 2a and b if it was also in weeks. 2b shows that KO at 21 weeks is the same size as WT at 16 weeks. This is not apparent in 2c.

-Thank you for the suggestion! We revised Supplementary Fig. 3a (the original Fig. 2c) by showing in weeks.

-In revised Fig. 2b and Supplementary Fig. 3a, we show two different experiments using different sets of mice. In Supplementary Fig. 3a, tumor growth was monitored for 30 days by measuring the tumor size every 5th day regardless of age. As a result, KANK1-WT^{PyMT} mice with fast-growing tumors were sacrificed before reaching the 16 weeks end point, and hence, they could not be included in Fig. 2b. On the other hand, some KANK1-KO^{PyMT} mice had late tumor onset, even close to the 21 weeks end point. They were also not included.

4. Line 183 - "interacting domain" - UC-Ank is not really a domain, but a region of unstructured adjacent to the Ank repeats.

-The text has been revised accordingly (see line 222 and line 227).

5. Should Supp Fig 4b and c have Upper and Bottom labels?

-The original Supp Fig.4b,c are now shown as revised Supplementary Fig. 6b,c. Supplementary Fig. 6b shows an Upper image section with the junctional signals, whereas Supplementary Fig. 6c shown the maximal signal projections from z-stack imaging to highlight the absence of Talins at cell-cell junctions.

6. Line 91. The justification for looking at KANK1 in breast cancer is not clear and does not correspond to data shown in fig.1. The description of the rationale for looking at KANK1 in breast cancer and could be improved.

-We describe the rationale in the revised introduction section (see line 60-64) and the revised result section (see line 109-131).

7. Fig 2f. The Y axis, # of positive cells/animal is unclear, is it in the mammary gland? Mammary epithelium?

-Yes, we quantified the signals in mammary epithelial tumor cells, which are identified by E-cadherin (E-cad) immune-fluorescence staining. The E-cad signal is shown in the revised Supplementary Fig. 4a-b and described in the Methods Section (see line 468-470).

Reviewer #5 (Remarks to the Author):

Reviewer #1 (Remarks to the Author)

In my previous comments, I asked the authors to carry out new *in vitro* and *in vivo* experiments to fully support their claim of causal connections between KANK1, NOS1AP, Scribble and TAZ (Fig. 8j).

In answer, the authors provided several new data from experiments carried out *in vitro*, showing the expected epistatic relationships between KANK1 and the other factors (although the effects appear rather weak, see the graphs on the right of Supp.Fig.11e).

Yet, the authors did not perform any of the *in vivo* experiments I requested in my points 1, 2, 3 and 5, saying it was because they lacked the necessary permissions. This is apparently true also for my point 3, in which I requested to monitor formation of metastatic nodules on the mice they already used for their initial submission. In that case, the authors have the permission to work with tumor-developing transgenic mice, but that permission does not allow mice reaching a stage in which rampant metastases would form.

I sympathize with the authors' struggle with the roadblocks posed by the EU against animal experimentation; nonetheless, I still think that the main claim of this MS needs some *in vivo* validation, perhaps through collaborations with some other laboratories outside EU. In particular, it would be important to validate that TAZ works as the downstream effector of KANK1 to foster tumor growth, showing that Hippo-resistant TAZ can rescue the effects of KANK1 loss. Without that, the claim of a new oncogenic pathway involved in the biology of breast cancer would be rather weak, as it would rely only on *in vitro* experiments showing weak effects. As results, the story would result poorly attractive for general readers and readers interested in cancer biology.

We appreciate that the reviewer sympathizes with our situation. It is important to demonstrate that the Hippo-resistant TAZ can reverse the effects of KANK1 loss, and we are really thankful that reviewer #1 raised this question. As pointed out in the previous point-by-point rebuttal, we have to deal with the EU laws on animal experiments and in addition, face a slow, bureaucratic evaluation process by the local, Bavarian government. It would take a year to get permission and at least an additional 6-12 months to perform the actual *in vivo* experiment.

We are also bound to the decision that tumor-bearing mice have to be killed as soon as tumors reach the size of up to '3 cm diameter in sum'. Since we are also interested to determine the metastasis rate of KANK1-WT^{PyMT} versus KANK1-KO^{PyMT} tumors, we will also apply for tumor experiments in FVB mice which seed cancer cells early into distant organs.

To satisfy the reviewer and ourselves regarding the Hippo-resistant TAZ for tumor cell growth, we decided to test the growth potential in 3D tumoroid cultures. Although 3D tumoroid cultures are not *in vivo* experiments, they closely mimic the *in vivo* conditions in organs or tumors and are therefore, widely used to substitute for *in vivo* experiments (PMID: 24912145 and 28520521). These experiments show that the Hippo-resistant TAZ can reverse the effects of KANK1 loss. The findings have been included in the revised manuscript.

Since it is important to mention the value of *in vivo* experiments in our revised manuscript, we discussed this limitation at the end of the Discussion section. Line 426-430: The limitation of the current study is that the significance of the stabilized TAZ^{SSAA} was not validated in a KANK1-KO background *in vivo*. Although we demonstrate that expression of TAZ^{SSAA} in MCF7^{KANK1-KO} tumoroids reversed the growth retardation of non-transduced MCF7^{KANK1-KO} tumoroids, xenografting TAZ^{SSAA}-expressing MCF7^{KANK1-KO} cells *in vivo* will constitute strong evidence for KANK1 inhibition as a novel strategy to curb breast cancer growth in patients.

Reviewer #2 (Remarks to the Author)

After carefully considering the authors' responses to both my comments and those of the other reviewers, I find the revisions satisfactory. The authors have adequately addressed the concerns raised in the initial review, and the manuscript has been significantly improved. I recommend its acceptance in Nature Communications without further revision.

Andreja Ambriović-Ristov

Thank you for the support.

Reviewer #3 (Remarks to the Author)

I am satisfied with the revision.

Thank you for the support.

Reviewer #4 (Remarks to the Author)

The authors have effectively addressed the points raised in the initial review and the revisions made to the manuscript are both thorough and satisfactory. The revisions have significantly enhanced the clarity of the manuscript.

However, it remains unclear from the data presented in Figure 7e and h that LATS activity is higher in KANK1-KO. While it's agreed that the ratio is increased based on normalized data (i.e. the proportion of LATS that is phosphorylated has increased), the total LATS activity appears unchanged. Nonetheless, this is not a major point and should not impede acceptance of an otherwise very solid manuscript.

Thank you for the support.

We mention in the revised Result section that the LATS activities (phosphorylation level) are increased relative to the total LATS levels (see line 298).

Minor point:

Fig1 B – Typo in “Subtype”

Thank you for making us aware of the mistakes in Fig. 1b and Supplementary Fig. 12b. The mistakes have been corrected in the revised manuscript.

Reviewer #5 (Remarks to the Author)
